# Cryo-EM structure of the sodium-driven chloride/bicarbonate exchanger NDCBE

Weiguang Wang[1,2], Kirill Tsirulnikov[1], Hristina R. Zhekova[3], Gülru Kayık[3], Hanif Muhammad Khan[3], Rustam Azimov[1], Natalia Abuladze[1], Liyo Kao[1], Debbie Newman[1], Sergei Yu. Noskov [3], Z. Hong Zhou [2,4], Alexander Pushkin[1] & Ira Kurtz[1,5 ✉]

SLC4 transporters play significant roles in pH regulation and cellular sodium transport. The previously solved structures of the outward facing (OF) conformation for AE1 (SLC4A1) and NBCe1 (SLC4A4) transporters revealed an identical overall fold despite their different transport modes (chloride/bicarbonate exchange versus sodium-carbonate cotransport). However, the exact mechanism determining the different transport modes in the SLC4 family remains unknown. In this work, we report the cryo-EM 3.4 Å structure of the OF conformation of NDCBE (SLC4A8), which shares transport properties with both AE1 and NBCe1 by mediating the electroneutral exchange of sodium-carbonate with chloride. This structure features a fully resolved extracellular loop 3 and well-defined densities corresponding to sodium and carbonate ions in the tentative substrate binding pocket. Further, we combine computational modeling with functional studies to unravel the molecular determinants involved in NDCBE and SLC4 transport.

[1] Department of Medicine, Division of Nephrology, David Geffen School of Medicine, University of California, Los Angeles, CA, USA. [2] Electron Imaging Center for Nanomachines, California NanoSystems Institute, University of California, Los Angeles, CA, USA. [3] Centre for Molecular Simulation, Department of Biological Sciences, University of Calgary, Calgary, Canada. [4] Department of Microbiology, Immunology and Molecular Genetics, University of California, Los Angeles, CA, USA. [5] Brain Research Institute, University of California, Los Angeles, CA, USA. ✉email: ikurtz@mednet.ucla.edu

NDCBE (SLC4A8) mediates Na[+]-driven Cl[−]/CO₃[2−] exchange and belongs to the Na[+]-dependent group of the solute carrier family 4 (SLC4) transporters[1,2]. Transporters of this group (SLC4A4, SLC4A5, SLC4A7, SLC4A10 and likely SLC4A9) are also involved in electrogenic or electroneutral Na[+]−bicarbonate(carbonate) cotransport or sodium dependent chloride/carbonate exchange[1,2]. Another group of SLC4 transporters, namely anion exchangers 1 (AE1, SLC4A1), 2 (AE2, SLC4A2) and 3 (AE3, SLC4A3), mediate Na[+]-independent electroneutral Cl[−]/HCO₃[−] exchange. Distinct from these transporters is SLC4A11 with the lowest homology to all other SLC4 family members, which transports NH₃−H[+] and H[+](OH[−])[3]. NDCBE exchanges one Na[+] and one CO₃[2−] ion for one Cl[−] ion[4,5]. Its transport mode has similar features with both anion exchangers and sodium bicarbonate(carbonate) cotransporters. NDCBE is highly expressed in all major regions of the brain, pituitary gland, and testis, and is also found in the trachea, thyroid, kidney, and pancreas[5]. Na[+]−driven Cl[−]/CO₃[2−] exchange activity plays an important role in pH regulation in specific cells throughout the brain, kidney and the rest of the body[5–8]. Knockout mice have abnormal regulation of Na[+] reabsorption in the kidney, and decreased neuronal excitability[6,7]. Chronic continuous hypoxia was shown to decrease NDCBE expression in mouse brains[8].

NDCBE like other SLC4 transporters is believed to predominantly form dimers[2,9–11]. SLC4 monomers consist of a large C-terminal transmembrane domain (TMD), involved in transport mediation, and a smaller N-terminal cytoplasmic domain (CD). The CD in SLC4 transporters is involved in interactions with various cytoplasmic and cytoskeletal proteins[1,2,12–16].

Structures of the TMD domains in the outward-facing (OF) conformation have been solved by X-ray crystallography to 3.5 Å for human AE1 (hAE1)[10] and by cryo-EM to 3.9 Å for human NBCe1 (hNBCe1)[11]. Both structures possess a similar 7 + 7 transmembrane segment (TM) inverted repeat fold and can be separated into core (TMs 1-4, 8-11) and gate (TMs 5-7, 12-14) domains. Their putative ion coordination area was suggested to be formed by the N-terminal ends of the short TM3 and TM10 (where the dipole moment, generated by the polar N-H groups may provide the necessary environment for energetic stabilization of anions[17]) and an electronegative residue (aspartate or glutamate) in TM8[10,11,17]. However, the structures did not resolve bound ions. The 7 + 7 inverted repeat fold has also been reported in a plant borate transporter[18] homologous to the SLC4A11 NH₃−H[+] transporter[3] that has a significantly lesser homology with other SLC4 family members, and several unrelated transporters in the SLC23 and SLC26 families[18–23] implying that it is not itself the main determinant of their function. Indeed, using functional mutagenesis measurements[11], we demonstrated that NBCe1 can develop Cl[−]-dependent transport if several NBCe1 residues in its putative ion coordination area are replaced with their AE1 analogs suggesting that the architecture of the binding pocket of an SLC4 transporter affects its transport mode. Solving additional SLC4 transporter structures is required to address the transport mode determinants and further expand our knowledge of the structural mechanisms involved in the SLC4 mediated ion transport. Here we performed structural characterization of NDCBE, whose transport properties have features present in NBCe1 and AE1. Our NDCBE structure provides the location of the ion binding sites in a SLC4 transporter and has fully resolved all loops including the largest extracellular loop 3 (EL3). The structural studies were followed by functional mutagenesis and computational modeling for further assessment of the ion behavior within the ion coordinating site of NDCBE. Our work identifies several protein residues involved in ion binding and provides insights into the structural basis of Na[+]-dependent transport in the SLC4 family.

## Results

**Overall structure of NDCBE.** Full-length rat NDCBE (rNDCBE) was expressed in HEK293 cells, purified as described in Methods and used for cryo-EM studies. The structure consists of TMD and CD with TMD solved at significantly higher resolution (Fig. 1, Supplementary Fig. 1). The linker between the TMD and CD is flexible and discontinuous and results in a less well-resolved CD due to the dynamic nature and various orientations of this region relative to the TMD. To improve the resolution, a subset of 380,776 particles was used for focused 3D classification and refinement on the TMD region and yielded a final reconstruction at 3.4 Å showing TMD and all loops including EL3 (Fig. 1a, b, Table 1). The high quality of the TMD enabled the generation of an accurate atomic model consisting of residues 451 to 1021 (Fig. 1b). We mainly focus on the structure of TMD in this report.

The cryo-EM map reveals that NDCBE is a dimer with each monomer comprising 14 TMs (TMs 1-14), five amphipathic helices (H1-2, H7-9), and a C-terminal cytoplasmic helix (H10), together with the loops connecting all these helices including the long EL3 (Fig. 1b, c). TMs 1-7 and TMs 8-14 are inverted structural repeats. The TM helices can be divided into two structural domains, the core domain (TMs 1-4 and TMs 8-11 with H1-2 and H7) and the gate domain (TMs 5-7 and TMs 12-14 with H8-9), which are separated by a cleft on the extracellular side of the protein (Fig. 1d), providing ion transport cavities between the core domain and the gate domain. The resolution of the core domain reaches 3.0 Å (Supplementary Figs. 1f and 2). The boundaries between the core and gate domains are well defined and clearly visible in top views (Fig. 1d). Helix H1 is located on the cytoplasmic side and is parallel to the plasma membrane before connecting to TM1. The loop–H2–loop connecting TMs 4 and 5 and loop–H7–loop connecting TMs 7 and 8 are across the core domain and gate domain on the cytoplasmic side and extracellular side respectively (Fig. 1c, d), thus linking the core domain to the gate domain.

A long EL3 (residues 621–694) located between TM5 and TM6, is a characteristic among the Na[+]-dependent SLC4 family members. EL3 in AE1 (Supplementary Fig. 3) and two other anion exchangers, AE2 and AE3[2], is significantly shorter. In the NDCBE structure, EL3 forms a relatively rigid domain-like structure consisting of four short α-helices (H3−H6) and one pair of anti-parallel β-strand (Fig. 1b–d). EL3 is stabilized by two intramolecular disulfide bonds, Cys636−Cys684 and Cys638−Cys672, as well as hydrogen bonds between the anti-parallel β-strand (Fig. 1d, e). The two disulfide bonds are important for the folded conformation of EL3. Mutations of each of the four highly conserved cysteine residues make EL3 highly accessible to protease digestion[24]. EL3 is also located at the dimeric interface between two subunits and involved in dimerization (Fig. 1f). Two asparagine residues, Asn646 and Asn666, have visible densities of likely covalently linked N-acetylglucosamine sugar (Fig. 1g, Supplementary Fig. 2).

**NDCBE dimerization.** Biochemical and biophysical data indicated that SLC4 transporters are dimers[2,9]. Consistently, the cryo-EM structure of NDCBE revealed its dimeric organization (Fig. 1f, Supplementary Fig. 4a–d). Both TMD and CD were involved in dimeric assembly. Because of the dynamic nature of CD, its resolution was not high enough to map the CD residues involved in dimerization. Dimerization in EL3 is supported by formation of hydrogen bonds between the carbonyl oxygen of

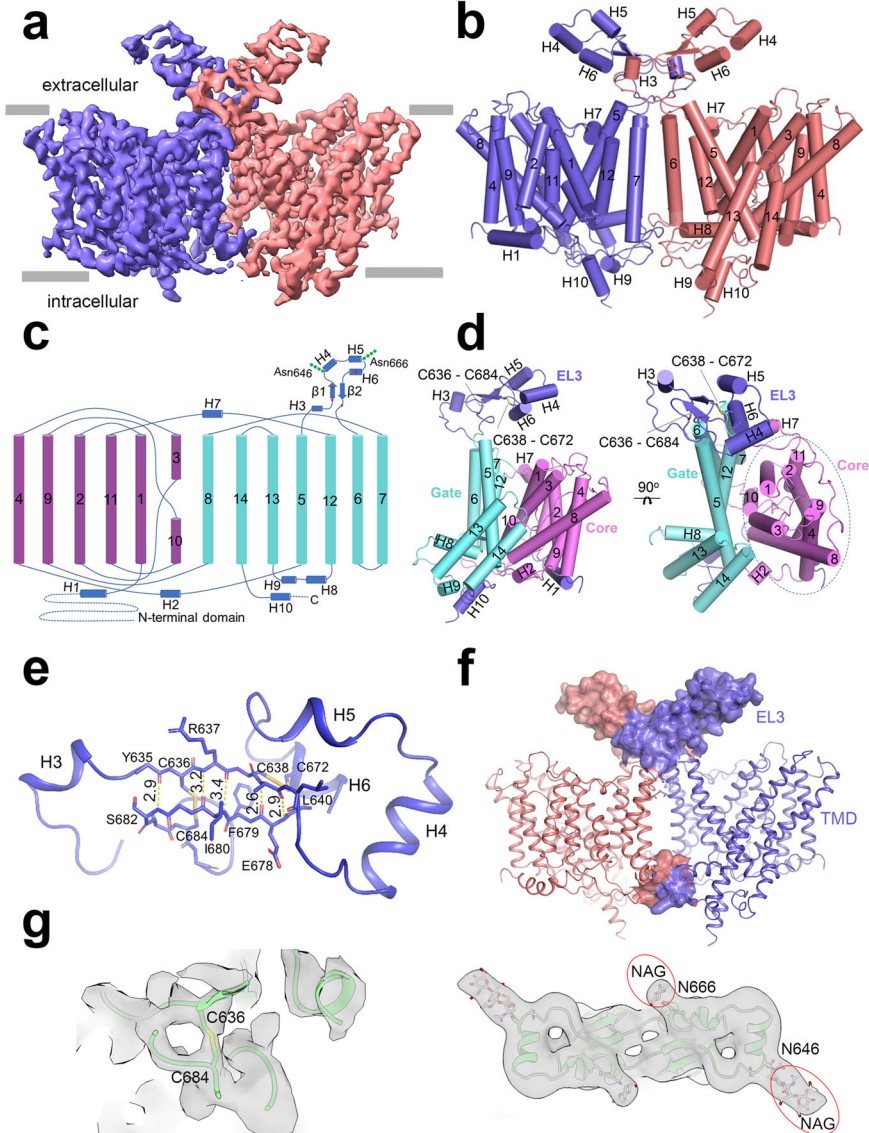

**Fig. 1 Overall structure of NDCBE TMD. a** Side view of a three-dimensional (3D) reconstruction of NDCBE with individually colored monomers. **b** Cartoon representation of NDCBE dimer. TMs 1-14 and helices H1-10 are shown as cylinders. **c** Topology and domain arrangement of the NDCBE monomer. Four highly conserved cysteines are labeled as red dot. The branched structures at Asn646 and Asn666 represent N-linked glycosylation. **d** Structure of NDCBE monomer in side (left) and extracellular (right) views. **e** Interactions contributing to EL3 assembly. **f** EL3 and TMD components involved in formation of the dimeric interface. **g** Cryo-EM densities showing the disulfide bond between Cys636 and Cys684 (left) and two N-glycosylation sites at Asn646 and Asn666 (right); the map was low-pass filtered to 5 Å for the two N-glycosylation sites.

Asp629/hydroxyl oxygen of Ser632 and the amidogen nitrogen of Val669 from the neighboring subunits (Supplementary Fig. 4b). In addition, the dimerization is likely strengthened by hydrophobic interactions between the side chains of Leu631 and Pro689 in EL3 of separate monomers. Residues Pro692, Phe696, Trp697, Ile700, Phe703 and Ile707 of TM6 interact with those of the neighboring TM6 (Supplementary Fig. 4c). Dimerization interactions are also formed between residues Met933, Pro934 and His937 of H8-loop-H9 and residues Arg719, Tyr720 and Pro722 of TM6-loop-TM7 from the neighboring subunit (Supplementary Fig. 4d).

**Ion binding sites of NDCBE adopting OF conformation.** TM3, TM10 and the antiparallel β-sheets preceding the two helices in the core domain have been proposed to contribute to substrate coordination in AE1[10] and NBCe1[11]. This region in our surface-rendered NDCBE atomic model presents as an extracellular-

facing cavity where solvent can access from the extracellular side to the ion coordination area, suggesting an outward-facing state (OF) (Fig. 2a). The 3.0 Å resolution cryo-EM map of the core region reveals two densities in this putative ion coordination area, approximately halfway across the membrane, likely corresponding to $Na^+$ and $CO_3^{2-}$ (Fig. 2b–d). $Na^+$ is coordinated by the hydroxyl oxygens of Thr804 and Thr847 and forms an ionic bond with the carboxyl group of Asp800, with coordination distances of around 2.4 Å, 2.6 Å and 2.6 Å, respectively. Since such local chemical environment explicitly favors a cation, we modeled a $Na^+$ at this site. The distances between Asp800, Thr847 and the $Na^+$ ion suggest that some of these residues may coordinate $Na^+$ indirectly via water molecules. The $CO_3^{2-}$ is 3.0 Å distant from $Na^+$. A $CO_3^{2-}$ oxygen is coordinated by the backbone amide group of Gly539 and also by the sidechain hydroxyl group of Thr538 with coordination distances of 4.0 Å and 2.9 Å, respectively. Another $CO_3^{2-}$ oxygen forms bonds with the sidechain

**Table 1 Data collection, refinement and validation statistics.**

| Data collection and processing | |
| --- | --- |
| Electron microscope | FEI Titan Krios |
| Voltage (kV) | 300 |
| Electron detector | Gatan K2 Summit |
| Magnification | 130,000 |
| Pixel size (Å) | 0.535 |
| Electron dose (e-/Å$^2$) | ~52 |
| Defocus range (μm) | −2.2 to −2.6 |
| Symmetry imposed | C2 |
| Map resolution (Å) | 3.4 |
| FSC threshold | 0.143 |
| Particles for final refinement | 381,293 |
| **Refinement** | |
| Software | COOT, EMBuilder, PHENIX |
| Model resolution (Å) | 3.8 |
| FSC threshold | 0.5 |
| Map sharpening B-factor (Å$^2$) | −143 |
| Model composition | |
| Non-hydrogen atoms | 9162 |
| Protein residues | 9040 |
| Ligands | 122 |
| R.m.s. deviations | |
| Bond lengths (Å) | 0.008 |
| Bonds angle (°) | 1.341 |
| **Validation** | |
| MolProbity score | 1.66 |
| EMRinger score | 2.10 |
| Clashscore | 1.4 |
| Poor rotamers (%) | 0 |
| Ramachandran plot statistics (%) | |
| Preferred (%) | 92.97 |
| Allowed (%) | 7.03 |
| Outlier (%) | 0.00 |

hydroxyl group of Thr538 and Na$^+$ with coordination distances of 2.8 Å and 3.0 Å, respectively (Fig. 2b). This chemical environment is suitable for anions and considering the OF state of the NDCBE structure and its transport properties[4], a CO$_3^{2-}$ was modeled at this site. The two ion binding sites were confirmed by molecular modeling and binding free energy calculations performed subsequently. The 3.0 Å distance between Na$^+$ and CO$_3^{2-}$ allows the two ions to coordinate each other strongly, thus achieving stoichiometric cotransport.

**Molecular modeling of ion binding**. The resolved TMD structure was used for a series of Site Identification by Ligand Competitive Saturation (SILCS) calculations to probe tentative locations of cation-selective and anion-selective pockets in the OF permeation cavity of NDCBE. The SILCS method combines Grand-Canonical Monte Carlo and MD protocols and uses a panel of organic fragments at saturating concentrations, which allows mapping of the most likely areas of protein-fragment coordination, potentially revealing binding pockets and permeation pathways[25]. The SILCS maps in the NDCBE permeation cavity for the acetate oxygen (green mesh) and methylammonium nitrogen (purple mesh), SILCS probes for anions and cations, respectively, are shown in Fig. 3a. There is a well-pronounced maximum in the excess cation density in the vicinity of Asp800, which corresponds to the binding site for Na$^+$ assigned from the cryo-EM map of NDCBE (Fig. 2c). The anion maps reveal the presence of potential anion binding regions at the center of the

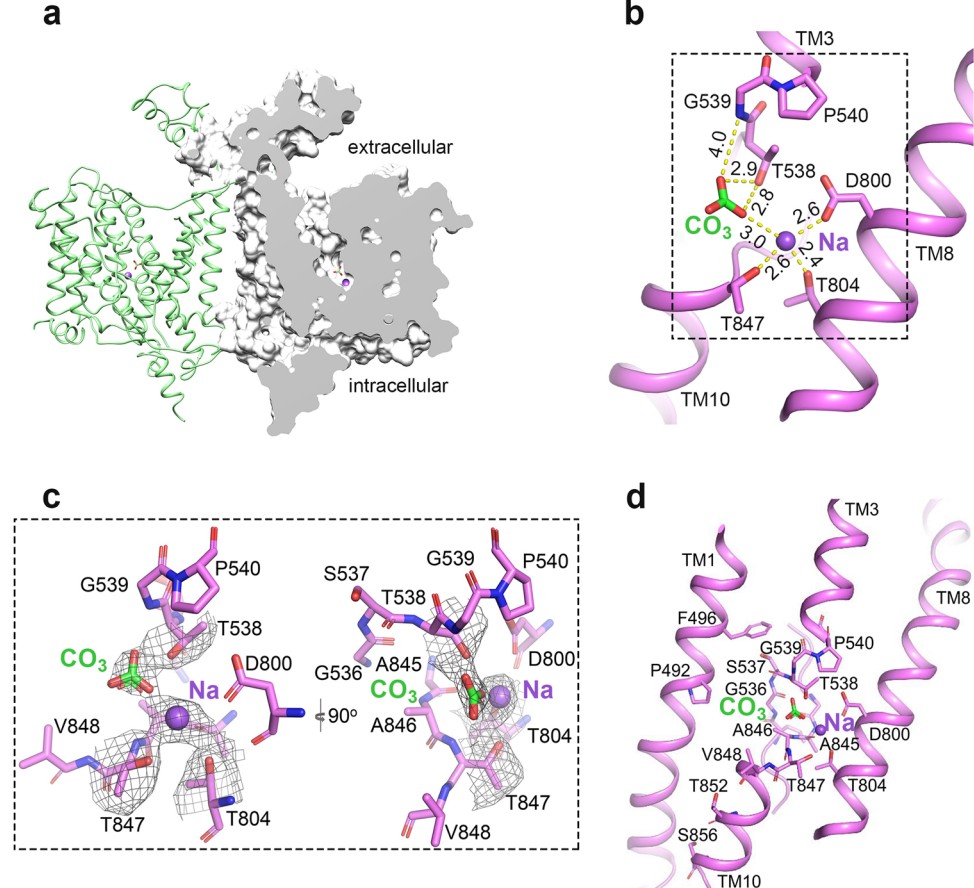

**Fig. 2 Two ion binding sites in OF NDCBE. a** An NDCBE monomer model presented as a surface (right) showing an extracellular-facing cavity. **b** The Na$^+$ and CO$_3^{2-}$ ions are coordinated at TM3, TM10 and TM8. Numbers show distances in Å between the ions and NDCBE coordinating atoms. **c** Magnified views of the Na$^+$ and CO$_3^{2-}$ binding sites in NDCBE. All cryo-EM densities are shown at the same contour level of 0.02σ. **d** Mapping of residues whose mutations significantly reduce NDCBE transport activity in and near the ion coordination area. Na$^+$ and CO$_3^{2-}$ are shown as a purple sphere and green-red sticks, respectively.

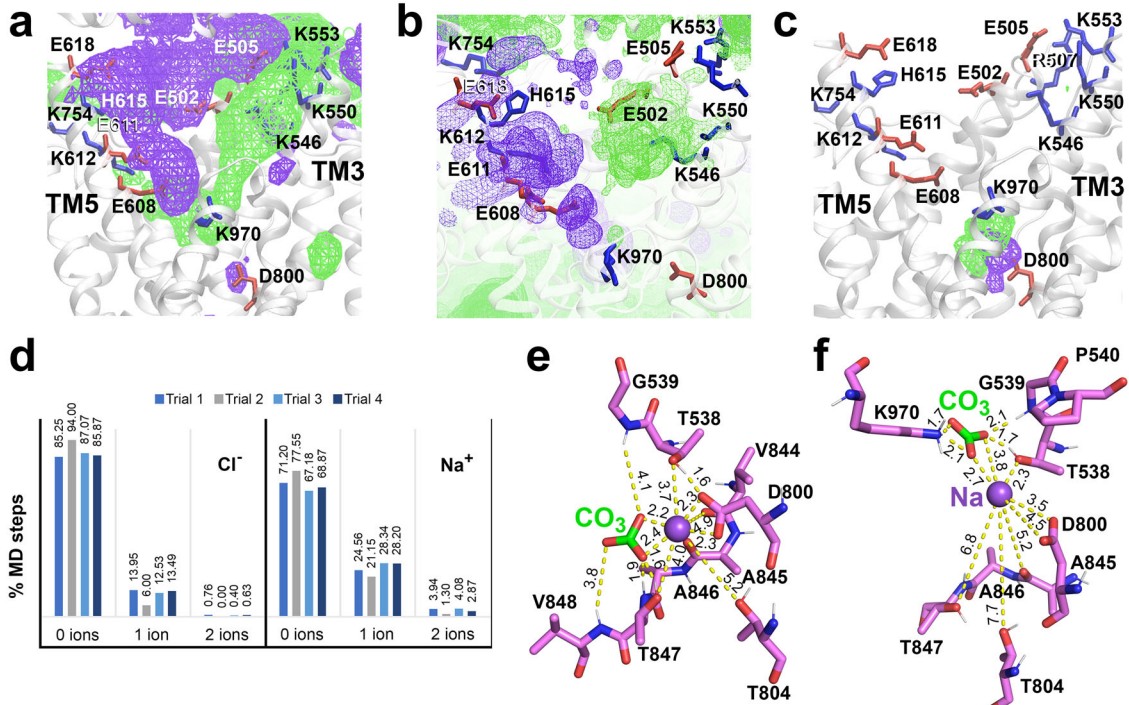

**Fig. 3 Computational modeling of NDCBE. a** SILCS maps demonstrating excess cation (purple mesh) and anion (green mesh) densities marking the cation and anion permeation pathways in the OF NDCBE cavity, respectively. The basic and acidic residues lining the permeation cavity are presented as blue and red sticks, respectively. **b** Close-up of $Na^+$ (purple mesh) and $Cl^-$ (green mesh) density maps in the permeation cavity of NDCBE obtained from 1 μs MD simulations of NDCBE initially loaded with $Na^+$–$CO_3^{2-}$ in the putative cryo-EM ion binding site. Maps from 4 different trajectories (Trials 1–4 in Supplementary Table 1) were overlapped on the figure. The ion maps were calculated for the portion of the MD trajectories after $Na^+$–$CO_3^{2-}$ have left the permeation cavity. **c** Sample density maps of $Na^+$ (purple mesh) and $CO_3^{2-}$ (green mesh), extracted from a 300 ns MD simulation (Trial 5 in Supplementary Table 1) of non-protonated NDCBE, loaded with a $Na^+$–$CO_3^{2-}$ pair. **d** $Cl^-$ and $Na^+$ permeation, calculated as percent of MD trajectory in which 0, 1, or 2 $Cl^-$ or $Na^+$ ions can be found within the permeation cavity of NDCBE from the four individual 1 μs MD simulations (Trials 1–4 in Supplementary Table 1). **e**, **f** MD equilibrated structures of sites S1 and S2, obtained from 300 ns MD simulation of NDCBE initially loaded with $Na^+$–$CO_3^{2-}$ in the putative cryo-EM ion binding site (Trial 5 in Supplementary Table 1). $Na^+$ and $CO_3^{2-}$ are shown as a purple sphere and green-red sticks, respectively.

protein marked by Thr538, Gly539, and Lys970, which is also in excellent agreement with the proposed cryo-EM anion binding site in NDCBE (Fig. 2c). In addition, the SILCS maps present two well-defined access pathways for the anions (along the lysine rich TM3) and $Na^+$ (along TM5 and the acidic residues Glu608, Glu611 and Glu618) in the permeation cavity of NDCBE (Fig. 3a).

**Putative ion binding sites in the permeation cavity of NDCBE from MD simulations.** To further elucidate the behavior of the SLC4 relevant physiological anions ($CO_3^{2-}$, $HCO_3^-$, $Cl^-$) and cations ($Na^+$) in the binding pocket of NDCBE and to probe the identity of the anionic substrate in the NDCBE cryo-EM map (Fig. 2c), we performed 1 μs and 300 ns all-atom MD simulations with different physiological $Na^+$/anion combinations (Supplementary Table 1). Since the reported OF NDCBE is not in occluded state as evidenced by the solvent accessible cavity (Fig. 2a) and unimpeded access to the protein center of SILCS fragments (Fig. 3a), over the course of the MD simulations, the ions dissociated from their initial positions and eventually left the binding pocket and the OF NDCBE cavity. The corresponding residence times in the binding pocket are listed in Supplementary Table 1 and the relevant ion density maps are presented in Fig. 3b, c. In the absence of $Na^+$, all studied substrates rapidly dissociate from the binding site and migrate from the permeation cavity into the external solution within the first ~10 ns of the simulation. In most simulations the $Na^+$-anion pair unbinds from its initial position and leaves the binding pocket at the same

time (see Supplementary Movie 1, Supplementary Figs. 5–7). The presence of a $Na^+$ ion stabilizes all tested anions in the binding pocket and leads to longer anion residence times, consistent with $Na^+$ dependent anion transport in NDCBE.

The 1 μs MD simulations allow direct assessment of the access of functionally relevant cations ($Na^+$) and anions ($Cl^-$) from the solution into the OF permeation cavity of NDCBE after $Na^+$-$CO_3^{2-}$ dissociation. Density maps for $Na^+$ and $Cl^-$ in the NDCBE permeation cavity obtained from these simulations are shown in Fig. 3b and generally overlap with the cation and anion permeation pathways extracted from the SILCS maps (Fig. 3a). Both $Na^+$ and $Cl^-$ from the external solution can enter the permeation cavity (Fig. 3d, Supplementary Figs. 5 and 7). At least 1 $Na^+$ is found in the permeation cavity in ~25% of the MD steps (about twice more frequently than 1 $Cl^-$).

Figure 3c presents sample ion densities of $CO_3^{2-}$ (green mesh) and $Na^+$ (purple mesh) extracted from one of the 300 ns MD simulations of NDCBE loaded with $Na^+$-$CO_3^{2-}$ (Supplementary Table 1, Trial 5). The densities reveal two putative ion binding sites, which we label S1 and S2. The MD equilibrated structures of sites S1 and S2 with bound $Na^+$-$CO_3^{2-}$, are shown in Fig. 3e, f. Site S1 (Thr538, Gly539, Asp800, Thr804, and Val844-Val848) squarely overlaps with the $Na^+$ and $CO_3^{2-}$ binding sites resolved by the cryo-EM map (Fig. 2c), with $Na^+$ position in agreement with the excess cation density from the SILCS map (Fig. 3a). Site S2 is located above site S1 in the OF NDCBE permeation cavity and involves residues Thr538-Pro540, Asp800 and Lys970. The $CO_3^{2-}$ density in site S1 overlaps with the excess acetate

oxygen density in this protein region, determined from the SILCS maps (Fig. 3a). The presence of a $CO_3^{2-}$ ion in this site may draw $Na^+$ ions from the solution, which then coordinate to Asp800 (the $Na^+$ binding area identified from our cryo-EM map) (see Supplementary Movie 2, Supplementary Fig. 7d). All simulations feature temporary coordination of the anion substrate in site S2 before the anions dissociate into the external solution (Supplementary Figs. 5 and 7). Short-lived migration from site S2 to site S1 of the $Na^+$-anion pair was also observed in some trajectories (see Supplementary Movie 3, Supplementary Fig. 7).

**Evidence for $Na^+$-dependent anion binding in site S1 from binding free energy calculations.** To further evaluate the substrate selectivity of the binding site (S1) revealed by the cryo-EM map, and by our SILCS and MD simulations, we performed binding free energy calculations for the ion loads presented in Supplementary Table 1. Supplementary Table 2 displays the absolute binding free energies for the different SLC4 physiological anions in the absence or presence of the co-permeant $Na^+$. It is evident that $Na^+$ enhances the anion binding to site S1 for all studied substrates (more negative $\Delta G_{bind}$ values in Supplementary Table 2). Binding of $Cl^-$ to site S1 in the absence of $Na^+$ is unfavorable (positive $\Delta G_{bind}$ of 6.2 kcal/mol), consistent with the short $Cl^-$ residence times observed in our MD simulations (Supplementary Table 1, Supplementary Fig. 7). $HCO_3^-$ and $CO_3^{2-}$ can still bind weakly to site S1 even in the absence of $Na^+$. The weak, but still favorable binding of $HCO_3^-$ and $CO_3^{2-}$ compared to $Cl^-$ is likely enabled by the nearby Lys970 from site S2 and by the better overlap of their trigonal planar shape with the backbone N-H bonds of residues Thr847 and Val848. Site S1 features strong selectivity for $CO_3^{2-}$ ($\Delta G_{bind}$ of −19.8 kcal/mol) over the other two studied anions when $Na^+$ is present. This finding, emphasizing the role of $Na^+$/solute coupling, is in agreement with the experimentally observed $Na^+$-$CO_3^{2-}$ transport and further supports the location of the $Na^+$ and $CO_3^{2-}$ sites proposed from the cryo-EM map (Fig. 2c).

**Functional mutagenesis.** Changes in the $Na^+$-driven base flux and membrane expression levels upon cysteine substitution for several residues in the vicinity of the putative binding sites of NDCBE are shown in Fig. 4 and Supplementary Figure 8. Mutations of residues Thr538-Pro540, Asp800, Ala845-Val848, and Lys970 found in the putative ion binding sites S1 and S2 identified from the cryo-EM map (Fig. 2c) and our MD simulations (Fig. 3e, f) reduced the NDCBE transport activity by ~20-60% relative to wild-type NDCBE (wt-NDCBE) (Fig. 4c). Combinations of these mutations decreased NDCBE function significantly below the function of individual mutants (Fig. 4c, d). Cysteine mutations of residues found in the vicinity of sites S1 and S2 (Fig. 2d), including Gly536 of TM3, Thr852 of TM10, Pro492 and Phe496 of TM1, also reduced NDCBE activity by ~20-60% (Fig. 4c). Given that all mutants have membrane expression levels not significantly different from wt-NDCBE (Supplementary Fig. 8), the observed $Na^+$-driven base flux decrease points to the involvement of the selected residues in ion coordination or as components in permeation pathways, as demonstrated from our cryo-EM and computational results (Figs. 2 and 3).

## Discussion
NDCBE is an electroneutral transporter of the SLC4 family that exchanges $Na^+$ and $CO_3^{2-}$ for $Cl^-$ and as such has both anion exchanger and cotransporter features. Therefore, comparison of the OF NDCBE structure (3.4 Å) with the previously solved OF structures of the anion exchanger hAE1 (3.5 Å) and the

$Na^+$-$CO_3^{2-}$ cotransporter hNBCe1 (3.9 Å) along with the resolved positions of the $Na^+$ and $CO_3^{2-}$ binding sites in the NDCBE cavity, can offer valuable insights into the intimate mechanisms governing $Na^+$-dependent anion transport in the SLC4 family. Similar to AE1[10], NBCe1[11], and several unrelated transporters[18–23], the NDCBE TMD possesses a $7 + 7$ TM inverted repeat fold, divided into core and gate domains (Fig. 1d). Interestingly, the available structures of AE1[10], NBCe1[11], and now NDCBE were all solved in the OF conformation.

The resolved structure of the complete glycosylated EL3 loop of NDCBE features a well-folded α/β domain stabilized by two disulfide bridges, located at the interface between the two monomers (Fig. 1b–f). The S-S bonds in NDCBE are formed between the first and fourth, and the second and third cysteine residues of EL3 whereas in NBCe1 the S-S bond formation involves the first and second, and third and fourth cysteine residues, as evidenced by previous biochemical and functional mutagenesis results[24]. It is currently unclear whether the differences in EL3 S-S patterns between NDCBE and NBCe1 are transporter specific. A member of the SLC26 transporter family, SLC26A2, with the same $7 + 7$ TM inverted repeat architecture, also has a glycosylation site in an extracellular loop stabilized by an S-S bond[26]. The identification of two densities in the EL3 NDCBE cryo-EM structure at Asn646 and Asn666 (Fig. 1g) indicates that two putative N-glycosylation sites contain polysaccharides. An interplay between the S-S bond formation and glycosylation in EL3 is involved in NBCe1 folding[24]. Whether such interplay between disulfide bond formation and N-glycosylation exists in NDCBE remains to be determined.

**NDCBE forms homo-dimers as do other transporters of the same architecture.** NDCBE is a homodimer similarly to AE1[10,27] and NBCe1[9,11]. A dimeric structure was also reported for several unrelated transporters featuring the $7 + 7$ TM inverted repeat fold as found in the resolved AE1, NBCe1 and NDCBE structures[18,20,21,26,28].

An important finding in this study is the structural involvement of EL3 in NDCBE dimerization (Fig. 1f, Supplementary Fig. 4). Given that the EL3 structure was not resolved in other SLC4 transporters, its role in their dimerization is uncertain, although based on the sequencing homology EL3 plays a potentially similar role in other $Na^+$-dependent SLC4 transporters. In addition, our data complement the reported dimerization of recombinant CD of NDCBE[29] supporting a potential involvement of NDCBE CD in dimerization similar to the dimerization of AE1 CD[15]. In SLC26A9 key interactions between the two monomers are thought to be mediated by the CD STAS domains[21]. The role of the CD domains in NDCBE dimerization and whether cholesterol is involved in stabilization of the NDCBE dimers as suggested for AE1[30] is as of yet unclear.

Our previous studies demonstrated that a dimer consisting of a wt-NBCe1 monomer and inactive mutant NBCe1 retains ~50% activity of NBCe1 indicating that the monomers in the NBCe1 dimer function independently of each other[9]. These experiments could not exclude the potential stimulation of monomer function by dimerization. Future experiments utilizing conditions preventing dimerization will shed light on this question. We have previously hypothesized that a potential advantage of dimerization is that the interaction of two gate domains can confer a greater stability in the plane of the membrane during the transport cycle[11].

**Ion binding area of NDCBE.** Structural comparison and sequence alignment of NDCBE, AE1 and NBCe1 demonstrate notable similarities and differences in their putative ion

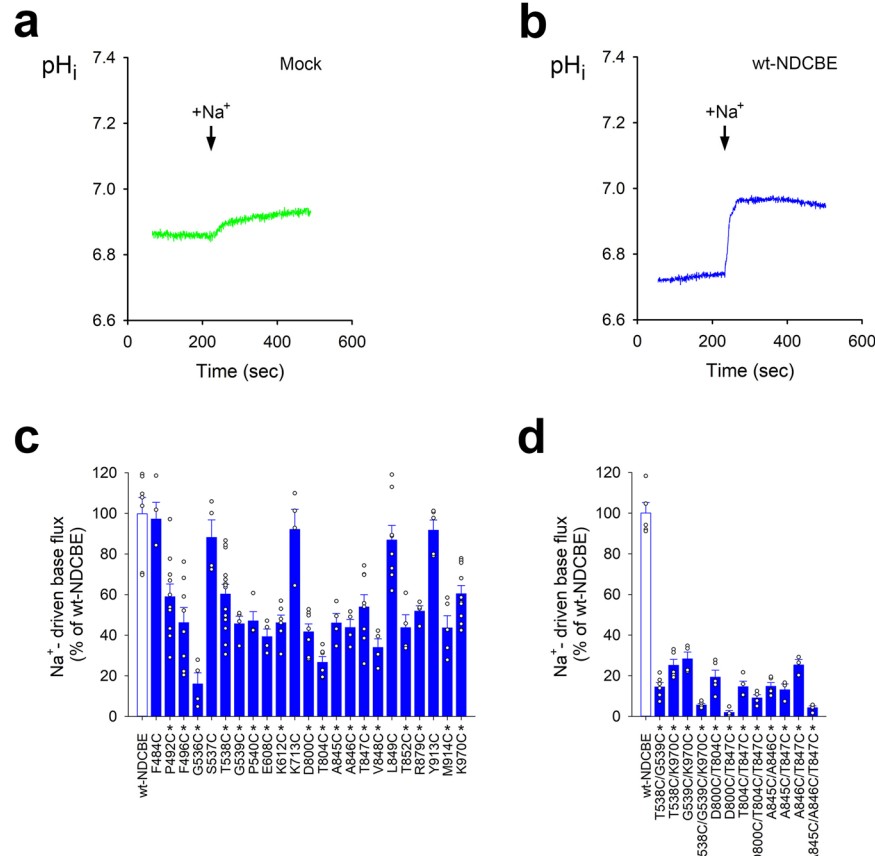

**Fig. 4 Functional studies of NDCBE mutants.** Typical functional traces of mock- (**a**) and wt-NDCBE (**b**) transfected HEK293 cells. **c** NDCBE wt ($n = 7$ biologically independent experiments) and single cysteine functional mutant data: F484C ($n = 4$, $p = 0.9997$); P492C ($n = 10$, $p < 0.001$); F496C ($n = 8$, $p < 0.0001$); G536C ($n = 4$, $p < 0.0001$); S537C ($n = 4$, $p = 0.9696$); T538C ($n = 14$, $p < 0.0001$); G539C ($n = 4$, $p < 0.0001$); P540C ($n = 4$, $p < 0.0001$); E608C ($n = 4$, $p < 0.0001$); K612C ($n = 6$, $p < 0.0001$); K713C ($n = 4$, $p = 0.9988$); D800C ($n = 7$, $p < 0.0001$); T804C ($n = 6$, $p < 0.0001$); A845C ($n = 4$, $p < 0.0001$); A846C ($n = 4$, $p < 0.0001$); T847C ($n = 8$, $p < 0.0001$); V848C ($n = 4$, $p < 0.0001$); L849C ($n = 8$, $p = 0.7539$); T852C ($n = 4$, $p < 0.0001$); R879C ($n = 4$, $p < 0.0001$); Y913C ($n = 5$, $p = 0.9948$); M914C ($n = 5$, $p < 0.0001$); and K970C ($n = 12$, $p < 0.0001$). Open circles represent individual data points. Source data are provided as a Source Data file. **d** NDCBE wt ($n = 5$ biologically independent experiments) and multiple cysteine functional mutant data: T538C/G539C ($n = 6$, $p < 0.0001$); T538C/K970C ($n = 5$, $p < 0.0001$); G539C/K970C ($n = 4$, $p < 0.0001$); T538C/G539C/K970C ($n = 5$, $p < 0.0001$); D800C/T804C ($n = 5$, $p < 0.0001$); D800C/T847C ($n = 4$, $p < 0.0001$); T804C/T847C ($n = 4$, $p < 0.0001$); D800C/T804C/T847C ($n = 4$, $p < 0.0001$); A845C/A846C ($n = 5$, $p < 0.0001$); A845C/T847C ($n = 3$, $p < 0.0001$); A846C/T847C ($n = 3$, $p < 0.0001$); and A845C/A846C/T847C ($n = 4$, $p < 0.0001$). One-way ANOVA and Dunnett's test were used to compare multiple study group means with wt-NDCBE. Statistically significant results differing from wt-NDCBE are depicted as mean ± SEM (*$p < 0.0001$). Open circles represent individual data points. Source data are provided as a Source Data file.

coordination areas and permeation cavities (Fig. 5, Supplementary Fig. 3)[10,11]. The binding pockets of NDBCE and NBCe1 are similar and differ from the binding pocket area of AE1 (Supplementary Fig. 3). A major difference is the presence in AE1 of a positively charged residue (Arg730) at the protein center instead of the nonpolar residues found in the $Na^+$-dependent NDCBE and NBCe1. In addition, there are significant differences between the three proteins in the charged residues of TM3 and TM5 (Supplementary Fig. 3), which line the putative ion permeation pathways as evidenced from our SILCS simulations (Fig. 3a).

**The residues of site S1 are among the main determinants of $Na^+$ dependence of SLC4 transport.** A combination of MD and SILCS simulations revealed the existence of two putative substrate binding sites, central site S1 and entry site S2, located in the large and well-hydrated OF ion-permeation cavity of NDCBE (Fig. 3c). Site S2 is positioned above site S1, at the bottom of the anion permeation pathway (Fig. 3a). It is temporarily occupied in all our MD simulations, after the ions dissociate from site S1 and likely serves as an attractor site (or "approach" site, according to

previous transport models of AE1[31,32], which captures ions from the permeation cavity and redirects them toward site S1 (see Supplementary Movies 2 and 3). The primary site S1 is at the center of the protein and likely corresponds to the previously proposed "alternating" site in AE1[31,32], where substrate binding initiates the conformational changes consistent with secondary transport. In AE1, the Arg730 residue is proposed to participate in anion binding in the area corresponding to site S1 in NDCBE[10]. The lack of an arginine residue in the NDCBE site S1 (the Arg730 analogs in NDCBE and NBCe1 are the non-polar Leu849 and Ile803, respectively, Fig. 5) therefore necessitates the involvement of a $Na^+$ ion, coordinated by Asp800, Thr804 and Thr847 in NDCBE, all of which decrease significantly the NDCBE transport upon cysteine substitution (Fig. 4). The hydrophobicity and size of Leu849 do not appear to play an important role in $Na^+$ coordination given that its replacement with the smaller and polar cysteine residue does not significantly affect NDCBE function (Fig. 4). Binding free energy calculations (Supplementary Table 2) show that $Na^+$ acts as an anchor of the anionic substrate in this site and leads to $Na^+$ dependent anion translo-

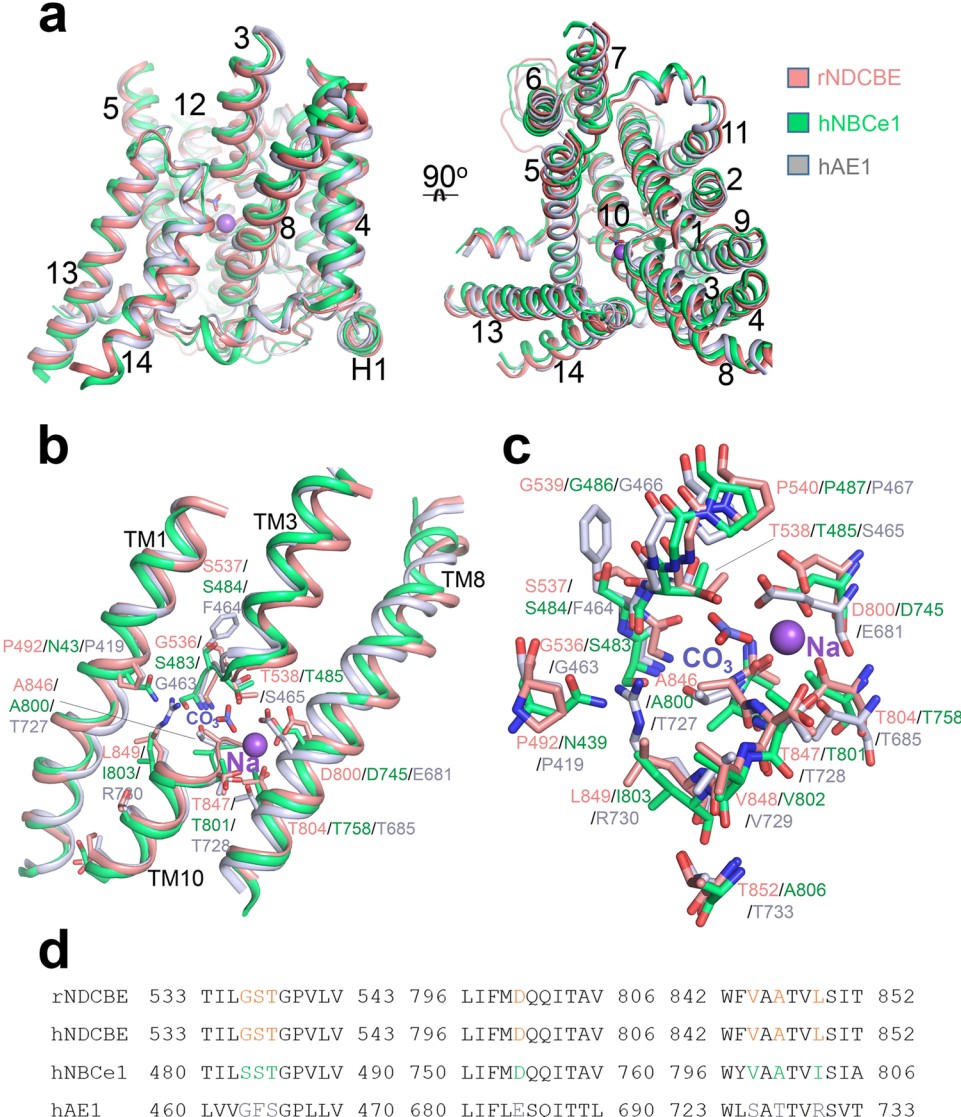

**Fig. 5 Structural comparisons of NDCBE with AE1 and NBCe1. a** The structural superimposition of NDCBE (salmon), AE1 (gray) and NBCe1 (green) in the TMD. For clarity, all loops including EL3 are not shown. **b** Comparison of NDCBE ion coordination residues with AE1 and NBCe1 residues located in or near the putative ion coordination area. **c** Magnified view of **b**. **d** Sequence alignment showing residues in and near the ion coordination sites. The Clustal Omega program[62] was used. $Na^+$ and $CO_3^{2-}$ are shown as a purple sphere and blue-red sticks respectively.

cation. Moreover, while $Na^+$ stabilizes different anions in site S1, the most favorable combination is the $Na^+-CO_3^{2-}$ ion pair, consistent with the functional evidence for $Na^+$-coupled $CO_3^{2-}$ transport in NDCBE[4,5]. Considering the high sequence similarity between NDCBE and NBCe1 in the area of site S1 (Fig. 5, Supplementary Fig. 3), this observation highlights the pivotal role of Asp800 in NDCBE and, potentially, Asp754 in NBCe1 (Fig. 5) for the $Na^+$-coupled acid-base transport in these two transporters. Indeed, substitution of Asp800 in NDCBE with cysteine in this study and Asp754 in NBCe1 with glutamate significantly impairs NDCBE and NBCe1 function[11] showing the specificity of an aspartate residue in this position in NDCBE and NBCe1. The smaller aspartate residue in NBCe1 and NDCBE may allow the accommodation of the larger $Na^+$ than $H^+$ in site S1.

Other notable non-polar/polar analogs from the ion coordination area with potential effect on $Na^+$-coupling include non-polar NDCBE Ala846, NBCe1 Ala800 and polar AE1 Thr727; polar NDCBE Ser537, NBCe1 Ser484 and non-polar AE1 Phe464; non-polar NDCBE Val844, NBCe1 Val798 and polar AE1 Ser725. These residues are in the area of site S1 determined from our

cryo-EM map and MD simulations. Residues Thr538 and Gly539 in NDCBE, which correspond to the NBCe1 residues involved in disease-causing mutations (T485S and G486R) are both engaged in $CO_3^{2-}$ coordination in our NDCBE structure (Figs. 2 and 3) and are sensitive to cysteine substitution (Fig. 4).

Surprisingly, site S1 does not bind $Cl^-$ effectively, as evidenced by our free energy calculations (Supplementary Fig. 2) and MD simulations (Supplementary Figs. 5 and 7). Previous functional studies indicate that NDCBE typically imports $Na^+$ and $CO_3^{2-}$ in exchange for $Cl^-$. Thus, the NDCBE selectivity for $Cl^-$ may be more pronounced in the IF binding pocket, which would necessitate the resolution of an IF state. The $Cl^-$ transport by NDCBE might also require additional considerations, such as protonation of residue Asp800 in site S1 (without subsequent NDCBE proton transport) which would remove the negative charge in the area and make binding of an anion possible without the involvement of a $Na^+$ ion. The pKa value of residue Asp800 evaluated by Propka is ~8.0 pKa units, which implies that it might easily protonate at physiological pH. Accordingly, AE1 mediates

$H^+$-$SO_4^{2-}$ cotransport, which is strongly dependent on Glu681 (the AE1 analog of Asp800) that is thought to be protonated[32].

**Transport mode can be potentially influenced by ion permeation and dynamics in the OF cavity of NDCBE.** The OF NDCBE permeation cavity is accessible to both $Na^+$ and $Cl^-$ ions (Fig. 3a–d). Our SILCS maps reveal distinct ion pathways for cations and anions in NDCBE, along the acidic residues in TM5 and the basic residues in TM3, respectively, where the amino-acid sequences of NDCBE, AE1, and NBCe1 show considerable differences (Fig. 5), which suggests that transport mode differences in SLC4 transporters may have an ion permeation component in addition to the energetics component underpinning the $Na^+$-dependent binding in site S1 (Supplementary Table 2). The OF NDCBE cavity is likely more permeable to $Na^+$ and less permeable to anions than the cavities of AE1 and NBCe1 due to the existence of two acidic residues on TM5 (Glu608 and Glu611, Supplementary Figs. 3, 5 and 7) and the presence of a His (His615) in the area of Lys562 in NBCe1 and Lys542 in AE1. The anion permeation pathway in AE1 and NBCe1 is expected to shift to TM5, due to the concentrated basic residues in this area and the fewer basic residues in TM3. Furthermore, residue Glu608 from TM5, which corresponds to Glu535 in AE1 and Asp555 in NBCe1, is in proximity to site S2 and is sensitive to cysteine substitution (transport decrease by ~60%, Fig. 4). Substitution of Asp555 to glutamate results in $Cl^-$ flux in NBCe1 and indicates that the acidic residues at this site in the SLC4 transporters play a role in transport mode determination[33]. A glutamate at this position may be required for $Cl^-$ binding and transport by the $Cl^-$ transporting SLC4 members[2].

In conclusion, we report here the near atomic resolution (3.4 Å) cryo-EM structure of a $Na^+$–$CO_3^{2-}$/$Cl^-$ exchanger (rNDCBE) that shares transport mode features with both AE1 ($Cl^-$/$HCO_3^-$ exchanger) and NBCe1 ($Na^+$–$CO_3^{2-}$ cotransporter). The structure captured densities corresponding to the bound ion substrates, $Na^+$ and $CO_3^{2-}$, in the ion coordination site located at the center of the protein, which was strongly supported by molecular modeling and functional mutagenesis. A second ion binding site, which controls the entry of ions in the protein center, was identified in the vicinity of the central binding site. A $Na^+$ ion, bound at an Asp residue in the center of the $Na^+$-dependent transporter NDCBE (and by sequence analogy, NBCe1), is necessary to anchor the anion in the binding pocket in the absence of a positively charged residue (such as Arg730 in $Na^+$-independent AE1) in this area. The charged residues lining TM3 and TM5, form a distinct ion permeation pathway from the extracellular solution to the OF permeation cavity of NDCBE. The identified two ion binding site organization as well as the mapped ion permeation pathways in NDCBE provide important insights into the transport mechanism differences observed among $Na^+$-coupled and $Na^+$-independent members of the SLC4 family.

## Methods

**Protein expression and purification.** For these studies we used rNDCBE that has a very high sequencing homology with human NDCBE (95.3% identity; Supplementary Fig. 3) given the significantly higher plasma membrane expression level of rNDCBE in HEK293 cells (ATCC) that was required for the structural studies. N-terminally Strep(II)-tagged wt-rNDCBE in the pTT vector (Addgene) was transfected into HEK293 cell monolayers using polyethylenimine. The cells were grown in Dulbecco's Modified Eagle's medium (Thermo Fisher Scientific) with 5% fetal bovine serum (Thermo Fisher Scientific) on 10-cm plates for ~24 h. The transfected cells from ~200 plates were washed and suspended in PBS, and then pelleted by centrifugation at 2000 *g* for 10 min. The cell pellets were solubilized with 2% Triton X-100 (Sigma-Aldrich) in buffer A (50 mM Tris-HCl, pH 7.5, 500 mM NaCl, both from Sigma-Aldrich) supplemented with complete protease inhibitor cocktail (Bimake) for 30 min. Detergent insoluble material was removed by centrifugation at 2000 × *g* for 30 min), and the supernatant was loaded onto the

5-ml StrepTrap HP column (GE Healthcare). The column was washed with buffer A containing 0.01% lauryl maltose neopentyl glycol (LMNG, Anatrace), and bound protein was eluted with the same buffer containing 2.5 mM D-desthiobiotin Sigma-Aldrich. Protein was further purified on a Superose 6 column (GE Healthcare) in 120 mM NaCl, 20 mM Tris-HCl, 20 mM NaHCO3, pH 7.4 with 0.01% LMNG. The peak fraction corresponding to dimeric NDCBE was collected and concentrated on an Ultracel 100 kDa MWCO (Amicon). As in our previous human NBCe1 purification scheme[11], DIDS was not used to lock NDCBE in the OF conformation, the CD was not removed via trypsin digestion, antibodies were not used for locking NDCBE in the OF conformation and the protein was not de-glycosylated as was done in the hAE1 purification and X-ray structural study[10].

**EM data acquisition.** The cryo-EM copper grids were prepared by applying 3.5 μl of NDCBE (~0.8 mg/ml) to a glow-discharged Quantifoil 200 mesh R1.2/1.3 grid and blotted with filter paper to remove excess sample for 2.5 s under 100% humidity at 4 °C before being plunged into liquid ethane using a FEI Vitrobot Mark IV. The grids were loaded into a FEI Titan Krios electron microscope operated at 300 kV. Micrographs (dose-fractioned movies) were acquired with a Gatan K2 Summit direct electron detection camera and SerialEM software[34] was used for automated data collection. A calibrated magnification of 130 kX was used for imaging with a pixel size of 0.535 Å. The defocus range was set from −2.2 μm to −2.6 μm. The dose rate on the camera was revised to ~7.9 e⁻ pixel⁻¹ s⁻¹ and the total exposure time was 8 s fractioned into 40 frames for each micrograph with 0.2 s exposure time for each frame. A total of 7545 micrographs were collected.

**Image processing.** The frame images of each micrograph were aligned and averaged for correction of beam-induced drift using MotionCor2[35]. The local motion within a micrograph was corrected using 5 × 5 patches. Two average images, with and without dose-weighting, were generated with 2x binning (final pixel size of 1.07 Å on the sample level) for further data processing. A total number of 7393 good micrographs were picked for image processing by visual inspection of the average images and power spectra with the drift correction in EMAN[36].

The defocus values of the micrographs were measured on the dose-unweighted average images by CTFFIND4[37]. The dose-weighted average images were used for particle picking and subsequent image processing. A total of 2,739,162 particles were automatically picked using RELION-3.0[38] and boxed out in 248 × 248 pixels. The extracted particles were subjected to three rounds of 2D classification and a total of 835,874 particles were selected for 3D classification (Supplementary Fig. 1c). The first round of 3D classification used an oval-shaped disk low-pass filtered to 60 Å as the reference. The particles were separated into six classes for 30 iterations. Three classes showing good TMD densities were selected to send to a focused 3D classification by using a mask encompassing the TMD and EL3. And one class showing more densities of the CD was further classified and 11,824 particles were selected for auto-refine. The final reconstruction of the full-length NDCBE is 18 Å by using the gold-standard FSC at 0.143 criterion. For the focused 3D classification, three classes showed good secondary structural features and their particles were selected for 3D auto-refine. After 3D refinement with C2 symmetry imposed and CTF refinement, a final 3D reconstruction from 380,776 particles yielded an EM map with a resolution of 3.4 Å by using the gold-standard FSC at 0.143 criterion. Half map 1 was used for post-processing and the atomic model was refined based on the full map. Angle distribution shows no preferred-orientation issue (Supplementary Fig. 1e). The local resolution was calculated by ResMap[39] using two cryo-EM maps to independently refine from halves of the data.

**Model building, refinement and validation.** De novo atomic model building was performed automatically by using EMBuilder[40] and manually built in Coot[41] based on the 3.4 Å resolution map. The high quality of the EM density map allowed us to construct a NDCBE model consisting resides from 451 to 1021. Amino acid assignment was achieved based on the clearly defined densities for bulky resides (phenylalanine, tryptophan, tyrosine, arginine and lysine) (Supplementary Fig. 2). The model was refined against the summed map using phenix.real_space_refine[42] with the secondary structure restraints and non-crystallography symmetry applied. To evaluate the signal of ions difference map, the experimental EM density map minus the simulated map from the refined model without ions was calculated. The distances between ions and coordinating residues were adjusted according to Harding[43]. The statistics for the model geometries was generated using Molprobity[44] (Table 1) and EMRinger[42]. Model and density figures were prepared in UCSF Chimera[45] or Pymol[46].

**Functional analysis**

*$Na^+$-driven $Cl^-$/$HCO3^-$ exchange transport assays.* Transport assays and flux measurements were performed as we previously described[11]. HEK293 cells were transfected with the wt-NDCBE and various NDCBE constructs, and 24 h later the transport assays were performed. On the day of study, the cells growing on 25.4 mm cover slips were initially bathed in the following $Na^+$-free, $Cl^-$-containing, bicarbonate-free solution: 140 mM tetramethylammonium chloride, 2.5 mM K2HPO4, 1 mM CaCl2, 1 mM MgCl2, 5 mM Hepes, pH 7.4, and 30 μM EIPA. After a steady state, NDCBE mediated $Na^+$-driven base flux in exchange for $Cl^-$ was initiated by switching to the following $Na^+$- and $Cl^-$-containing solution: 115 mM NaCl, 2.5 mM

$K_2HPO_4$, 1 mM $CaCl_2$, 1 mM $MgCl_2$, 24 mM $NaHCO_3^-$, 5% $CO_2$, pH 7.4, and 30 μM EIPA. The results of the functional measurements are presented in Fig. 4.

*Sulfo-NHS-SS-biotin plasma membrane labeling.* To determine the plasma membrane expression of the various constructs used in this study, Sulfo-NHS-SS-biotin was used. HEK293 cells expressing the various constructs 24 h following transfection were washed at room temperature with PBS at pH 8.0, and were then incubated at 4 °C (30 min; pH 8.0) with 1.1 mM sulfo-NHS-SS-biotin (Thermo Fisher Scientific). To stop the reaction, 50 mM Tris buffer at 4 °C (140 mM NaCl, pH 8.0) was used. After collecting the cells and washing with PBS, the cells were lysed with a solution containing 150 mM NaCl, 0.5% sodium deoxycholate (Thermo Fisher Scientific), 1% (vol/vol) Igepal (Sigma-Aldrich), 10 mM Tris-HCl, 5 mM EDTA (Sigma-Aldrich), pH 7.5, with protease inhibitors (Roche Life Sciences) on ice. After centrifugation at 20,000 g (4 °C) for 10 min, the insoluble fraction was pelleted and the supernatant (containing> 90% of the plasma membrane protein fraction) was collected and incubated at 4 °C for 4 h with 50 μl streptavidin-agarose resin (Thermo Fisher Scientific) on a rotating shaker. To elute bound proteins, the resin was pelleted by brief centrifugation and then washed with lysis buffer (at 60 °C) for 5 min with 2xSDS buffer containing 2% 2-mercaptoethanol (EMD Millipore). For detection of the lysate, the cells were lysed in lysis buffer containing 10 mM Tris-HCl, pH 7.5, 5 mM EDTA (Sigma-Aldrich), 1% (vol/vol) Igepal (Sigma-Aldrich), 150 mM NaCl and 0.5% sodium deoxycholate (Thermo Fisher Scientific), and NDCBE was pulled down using the rabbit polyclonal NDCBE-1rantibody (1:1000 dilution) against the C-terminal sequence LSINSGNTKEKSPFN that is identical in rat, mouse and human NDCBE[47]. The results are shown in Supplementary Fig. 8. The plasma membrane expression levels of all NDCBE mutants were not statistically different from that of wt-NDCBE (One-way ANOVA).

*SDS–PAGE and immunoblotting.* The protein samples were initially resolved on 7.5% polyacrylamide gels and then transferred to polyvinylidene difluoride (PVDF) membranes (GE HealthCare). The expression levels of the pulled down biotinylated proteins and whole cell lysates were assessed by probing the blots with our NDCBE-1r antibody in TBSTM buffer containing 137 mM NaCl, 20 mM Tris-HCl, pH 7.5, 0.1% (vol/vol) Tween 20 and 5% (w/vol) nonfat milk. After 1 h incubation at room temperature, the blots were washed with TBST and then probed at room temperature for 1 h with horseradish peroxidase (HRP)-conjugated AffiniPure Mouse Anti-Rabbit IgG (H + L) (Jackson ImmunoResearch Laboratories, Inc.) at 1:10,000 dilution in TBSTM buffer. The blots were washed with TBST and signals were detected with ECL Western Blotting Detection Reagent (GE HealthCare).

### Computational modeling

*SILCS calculation.* SILCS Software (Site Identification by Ligand Saturation, version 2020.2)[25] was used to map the potential $Na^+$, $Cl^-$ and $CO_3^{2-}$ ions binding sites and permeation pathways in NDCBE. Gromacs MD 2018[48] and CGenFF 2.3.0[49] with the Charmm36 and CGenFF force fields[50,51] and LINCS constraints were used for the all-atom simulations. Lennard-Jones interactions were handled by the Verlet cut-off algorithm with Force-switch option at 0.5–0.8 Å cut-off range. Columbic interactions were treated with the Particle Mesh Ewald algorithm using a 0.8 Å cutt-off. SILCS simulations were initiated on a NDCBE monomer embedded in a membrane bilayer (120:120 Å POPC-cholesterol mixture of 9:1 composition, solvated with TIP3P waters) using protocol described by Mousaei et al.[52].

Ten discrete systems were prepared after random distribution of water and fragment molecules. The molarity of each fragment and the water were set to 0.25 and 55 M, respectively. The systems were minimized and equilibrated following a six-step equilibration MD protocol where harmonic force constants applied to the protein backbone, protein sidechain and lipid heavy atoms were gradually decreased. Subsequently, 5 ns MD simulations were run with 50.208 kJ × mol$^{-1}$ × nm$^{-2}$ harmonic force constant imposed on the $C_\alpha$ atoms of the proteins using a protocol described previously[52]. Afterwards, 25 sequential GCMC computations were done. Each stage consisted of 200,000 MC steps with four types of plausible scenarios for water and probe molecules: insertion, deletion, translation and rotation. The movement probability was determined according to Metropolis criteria and depends on i) the target concentration of the solute ii) its excess chemical potential iii) energy change of the systems between two MC steps. Subsequently, 100 steps of GCMC/MD hybrid simulations were started where an MD step was inserted between each of the two GCMC steps for better conformational sampling. The MD portion included 5,000 steps of steepest descent minimization, followed by 100 ps of equilibration and 1 ns production run in NPaT ensemble (controlled with Nose-Hoover and semi-isotropic Parrinello-Rahman schemes) with a 2-fs time step, for a total simulation time of 1 μs (10 × 100 ns) at 300°K and 1 bar. From the simulation data, occupancy maps were generated to bin the selected solute atom types in a 1 Å$^3$ voxel volume spanning the protein. Then, these occupancy maps were converted into GFE maps (grid free energy maps) via a Boltzmann-based transformation[53]. Ultimately, GFE energy values were attained in the voxels that unravel the protein binding affinity sites. In the current work, all overlap coefficients that correspond to the relevant fragments are above 0.70 which indicates a satisfactory convergence of the simulations.

*Molecular dynamics simulations.* Before performing the MD simulations, the cryo-EM structure was refined further with the MDFF protocol as described previously[54,55]. CHARMM36m force field was used for the refinement[56].

The structure was minimized for 3 ns with a timestep of 1 fs. Secondary structure and chirality restraints were employed during the fitting. The simulation temperature was set to 303 K. A cut-off distance of 12 Å was used for nonbonded force evaluation with switch distance set to 10 Å. Multiple time step algorithm was applied to the van der Waals interactions (evaluated every 2 fs) and electrostatic interactions (evaluated every 4 fs) for all MDFF simulations. The MDFF–refined structures were selected to build protein-membrane systems for all-atom MD simulations.

MD simulations were performed for a number of NDCBE monomers with different ion loads (Supplementary Table 1). CHARMM-GUI Membrane Builder protocol was used to build and equilibrate all systems[57]. Each protein (Supplementary Table 1) was embedded in a POPC lipid bilayer (157 And 151 POPC molecules in upper and lower leaflet, respectively) in a periodic box of 113.10 × 113.10 × 130.52 Å dimension with 20 Å layers of water on both sides of the membrane and 150 mM NaCl. The initial position of the ions was guided by the cryo-EM density corresponding to the putative ion binding site in the 3.0 Å TMD cryo-EM map of NDCBE overlapping with the corresponding SILCS fragment densities maps. The equilibrated systems were subjected to 300 ns production runs in semi-iso-thermal-isobaric (NPaT) conditions with NAMD 2.12[58] and the CHARMM force field (CHARMM36m for proteins, CHARMM36 for lipids, TIP3P model for water and the available CGenFF parameters for $HCO_3^-$ and $CO_3^{2-}$)[51,56].

The production simulations were performed at 310.15 K and 1 atm set semi-isotropically. Particle Mesh Ewald was used for the long-range electrostatic interactions and the non-bonded interactions were cut-off and switched off at 12 and 10 Å, respectively. Four 1 μs-long MD simulations were performed for NDCBE monomers with a $Na^+$-$CO_3^{2-}$ pair in the putative cryo-EM binding site using the Anton2 computational platform and the protocol described previously[59]. The four replicas were prepared as described above and underwent 20 ns unconstrained MD equilibration before the 1 μs production runs, leading to different starting $Na^+$-$CO_3^{2-}$ positions in the S1/S2 binding pocket. The TMD structure remained stable in all 1 μs-long MD trajectories as evidenced by a plateau in time-traces of $C_\alpha$ Root-Mean-Square-Deviation relative to the starting structure (Supplementary Fig. 9). In-house scripts were used for trajectory analysis and evaluation of ion coordination and permeation. Ion density maps were generated with the VolMap tool of VMD 1.9.3[60].

### Absolute binding free energy calculations.

The absolute binding free energies for all ligands ($Cl^-$, $HCO_3^-$ and $CO_3^{2-}$) in site S1 of NDCBE, in the presence or absence of $Na^+$ (Supplementary Table 2), were calculated using the latest CHARMM-GUI Absolute Free Energy tool[61] and NAMD 2.15[58] with standard CHARMM36 and CGenFF parameters[50,51] for all binding and solvation simulations. FEP calculations were combined with the λ-Replica-Exchange Molecular Dynamics protocol to enhance sampling as described by Kim et al.[61]. The FEP alchemical transformation was done in 32 linearly-spaced windows with λ values from 0 to 1. All FEP windows were launched together and run concurrently by the multiple-partition module of charm + +/NAMD with the conventional Metropolis–Hastings exchange criterion for exchanges between different windows. Simple overlap-sampling estimator was used afterwards to post-process the potential energies of each replica–exchange. Two coupling parameters were used for Lennard-Jones and electrostatic interactions treatment. For all substrates bound to the complex a distance restraint was applied to restrain the ligand position in the $Na^+$-decoupled states[61]. The MD protocol was identical to that used for NAMD equilibration of production runs. All solute hydration calculations were performed for a single solute restrained at the center of the cubic simulation box with the dimensions of 52 Å × 52 Å × 52 Å containing 4,715 TIP3P water molecules as described by Kim et al.[61]. For each of the absolute $G_{hydr}$ calculations, FEP/λ REMD simulations were performed for 5 ns per window with the last 4 ns used for free energy calculations. For $G_{site}$ FEP/λ REMD was performed for 7 ns with the last 5 ns used to calculate the final free energy values. The standard errors in the calculated free energy values were within 0.5 kcal/mol.

**Reporting summary.** Further information on research design is available in the Nature Research Reporting Summary linked to this article.

## Data availability

The final cryo-EM density map of NDCBE has been deposited to the Electron Microscopy DataBank (EMDB) under the accession code EMD-24683. The final atomic model was deposited into the Protein Data Bank (PDB) under the accession code 7RTM. The data are provided in the Supplementary Information files, or can be obtained from the corresponding author upon request.. Initial and final steps from the MD trajectories are provided as Supplementary Data 1. Source data are provided with this paper.

## Code availability

The in-house scripts used for trajectory analysis are available at https://github.com/hzhekova/Scripts_for_NatComm.

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

## Acknowledgements

I.K. was supported by funds from the National Institutes of Health (NIH R01 DK077162), the Allan Smidt Charitable Fund, the Ralph Block Family foundation, and the Factor Family Foundation. Work in SYN lab was supported by the NIH (R01 DK077162) and partially by the Natural Sciences and Engineering Research Council of Canada (NSERC grant No. RGPIN-2021-02439). HZ was supported by NIH (R01 DK077162). MD simulations were performed on the Canada Foundation for Innovation supported GladOS cluster at the University of Calgary and on the West-Grid/Compute Canada clusters under Research Allocation Award to SYN. Part of the production simulations were performed on the Anton2 computer provided by the Pittsburgh Supercomputing Center (PSC) and DE Shaw Research through Grant R01GM116961 from the NIH. We acknowledge the Electron Imaging Center for NanoMachines for the use of their instruments supported by the University of California at Los Angeles and instrumentation grants from both the NIH (1S10RR23057 and 1U24GM116792) and the NSF (DBI-1338135 and DMR-1548924). The content is solely the responsibility of the authors and does not necessarily represent the official views of the National Institutes of Health.

## Author contributions

W.W. and Z.H.Z. collected and analyzed cryo-EM data. Z.H.Z. supervised cryo-EM experiments. L.K. and D.N. expressed NDCBE in HEK293 cells, and K.T. performed purification and biochemical characterization of NDCBE. L.K., R.A, D.N. and N.A. performed functional mutagenesis experiments. H.R.Z., G.K., H.M.K. and S.Y.N designed and performed computational modeling. W.W., A.P., H.R.Z., S.Y.N., Z.H.Z. and I.K. wrote the first version of the manuscript. All authors contributed to the critical revision of the manuscript and approved the final version.

## Competing interests

The authors declare no competing interests.
