## [Peer Review File · Nature Communications]

REVIEWER COMMENTS

Reviewer #1 (Remarks to the Author):

Nature Comms 21-09497

The exciting paper from the Kurtz lab provides new insights into the structure and mode of operation of the sodium-dependent chloride/bicarbonate exchanger (NDCBE), a member of the SLC4 solute carrier family. As expected, NDCBE has an identical 14 transmembrane (TM) dimeric structure to other members of SLC4 family, featuring a 7 + 7 TM inverted topology, complex folded core and gate domains and two short opposing TM 3 and 10 helices. A novel feature revealed by this 3.5 Å cryo-EM structure is the presence of a tightly-folded glycosylated extra-cellular loop (EL3) joining TM segments 5 and 6 at the dimer interface. A second important insight is the identification of the sodium-carbonate ion pair central binding site. A third feature discovered in the structure is a more distal carbonate gathering site. The role of residues involved in substrate binding and translocation were confirmed by site-directed mutagenesis and computer modelling studies, further strengthen this fine structural study.

Comments

1. Add SLC4A8 to the title

2. Line 78 Include comprehensive review on SLC4A1 in reference list:
(<https://pubmed.ncbi.nlm.nih.gov/27058983/>)

3. Line 79. Mention 7+7 fold commonly found in SLC4, SLC23 and SLC26 families with reference to:
(<https://pubmed.ncbi.nlm.nih.gov/27865089/>)

4. Lines 87/164/282 Important to mention in the paper that the OF state was attained without use of inhibitors (or stabilizing antibody) as in the SLC4A1 structure. Line 282: "Interestingly, the finding that the structures of AE1, NBCe1, and now NDCBE were solved in the OF conformation suggests that this is the most stable conformation of the SLC4 transporters." This comparison is not valid because the structure of AE1 referenced may not be representative of its native conformation since it is 1) a crystal structure 2) of cleaved membrane domain 3) deglycosylated 4) bound to an antibody fragment and 5) labelled with H2DIDS.

5. Line 108 Understand that the low resolution of the cytoplasmic domain precluded modelling. Were the authors able to determine whether the membrane and cytoplasmic domain had a cross-over domain-swap structure like SLC26A9, which is important for dimer formation:
(<https://elifesciences.org/articles/46986>)?

6. Line 174 The carbonate is located precisely between the N-terminal ends of TM3 and TM10. Some mention should be made of the possible role of the positive helix dipole in stabilizing anions at this site (<https://pubmed.ncbi.nlm.nih.gov/27058983/>).

7. Line 179 The sodium and carbonate likely form a stabilized ion pair in the S1 site accounting for the 1:1 stoichiometry. Transporters with other substrates stoichiometries may have evolved to employ different residues. E.g., The smaller Asp versus Glu in SLC4A1 can accommodate Na⁺ versus H⁺ in SLC4A1. Mention that SLC4A1 can act as a proton-sulfate co-transporter: (<https://pubmed.ncbi.nlm.nih.gov/1352774/>).

8. Line 302 Some brief mention of the role of N-glycosylation should be made. Interestingly, the SLC26A2 family member also has a disulfide-stabilized tightly folded extracellular N-glycosylated domain (<https://pubmed.ncbi.nlm.nih.gov/28941661/>).

9. Line 315 While the membrane domain of SLC4A1 is a stable dimer, mention the potential role of the dimeric cytoplasmic domain, including the cross-over domain swap in SLC26A9 in Ref 21. Also mention the potential role of lipids like cholesterol in stabilizing the dimer interface: (<https://www.sciencedirect.com/science/article/pii/S0006349519307465>).

10. Line 324 Fig. 5 a-d is referencing the wrong figure (should be figure 6)

11. Line 336 Leu 849 is not included in some of figures other than supplemental. It appears that a hydrophobic residue at this site is a feature (Fig. 6d) of sodium-dependent transporters, while exchanger have a larger and positively-charged Arg at this site. Does solvent occupy the space or is there a structural adjustment in this area.

12. Thirty some references (22-59) referring to various methodological details seems excessive. Perhaps reference could be made to earlier work using similar computational methods.

13. Data acquisition, lines 453-455. Based on the reported dose rate on the camera (9 e⁻ pixel⁻¹ s⁻¹), the total exposure time (8s) and reported pixel size (0.535 Å), the calculated total electron exposure is 251.55 e⁻/Å² (9 * 8 / (0.535²)), which is inconsistent with that reported in Table 1 (52 e⁻/Å²).

14. Clashscore of 16.3 reported in Table 1 is high, indicating a large number of clashes in the model. In our experience, automated refinement using PHENIX in this resolution range is not sufficient to resolve these issues. We recommend manual correction of these issues, or automated correction using Rosetta FastRelax or ISOLDE. A clashscore of approx. 1-2 is certainly achievable at this resolution.

15. The distances reported in Figure 3b between T538 and the carbonate anion are 2.2 and 2.4 Å, shorter than the 2.8 Å typically found in protein structures. In contrast, the distances between D800, T847, T804 and the sodium ion are greater than expected for sodium coordination (normally 2.42 Å, MM Harding Acta Cryst. (2002). D58, 872-874). Some of these residues could be indirectly coordinating the sodium ion by coordinating water molecules (which may not be visible in the map at 3.0 Å resolution). The distances described in this figure may change following refinement with other software packages described above.

16. The density for D800 in Figure 3c is not shown. Was this omitted or is the density not strong enough (due to radiation damage, which affects negatively charged residues)?
17. Orientation distribution should be shown in Figure S1.
18. EMRinger test for side-chain (available in Phenix) should be used, and results reported in Table 1.

Reviewer #2 (Remarks to the Author):

The manuscript by Wang and colleagues reported the structure of rat NDCBE (SLC4A8), a Na⁺-driven Cl⁻/CO₃²⁻ exchanger at near-atomic resolution using single-particle cryo-EM. Similar to previously determined structures in SLC4 family, NDCBE forms a dimer and the overall structure shows an outward facing conformation. Based on structural analysis, the putative ion (Na⁺, CO₃²⁻) binding sites have been identified and further confirmed by molecular dynamic simulation and mutagenesis.

One major point of this manuscript is identification of the ion binding sites for the first time in SLC4 family. However, the figure showing in the current version (fig. 3C) is difficult to evaluate the quality. The authors can calculate the difference map, i.e. experimental EM density map minus simulated map from refined atomic model without interested ion(s). The omit density can be used to evaluate the signal of the built ions. Also, the modeled Na⁺ and CO₃²⁻ are very close, could corresponding ion densities be accurately assigned confidently?

In line 170 of page 5, the authors claimed they modeled Na⁺ because that's the only cation they used in buffer. The possibility of endogenous ions binding to NDCBE tightly from protein expression can't be excluded.

In line 281 of page 9, the authors made conclusion that "... NDCBE were solved in the OF conformation suggests that this is the most stable conformation of the SLC4 transporters". Although current structures available in SLC4 family all present an OF conformation, this statement needs more evidence to support.

In line 320-322 of page 10, the authors said "Therefore, the transport mode of a SLC4 transporter is determined by a small number of residues in/near the ion coordination area rather than the overall 3D fold." First, even this statement is correct, 3D structure of the protein is the fundamental basis

that allows the arrangement of these key residues in a way to play their important roles. Secondly, current evidence is not solid and strong to support this statement. The author need experimental data to show that not only functions are switched by replacing key residues, but other features for transporter and exchangers in SLC4 family, like ions selectivity and specificity, are also maintained as well as the wide type protein.

The manuscript should be revised to make a more smooth manuscript. For example, description of the cryo-EM analysis (the first part of Results section in page 3) contains technical details that are redundant with the one in methods section. This part could be integrated into other structural analysis part, like overall structure (Page 4) and/or NDCBE dimerization (Page 5) section. Similar redundant descriptions is found in the discussion part that already exist in Results part. Also, the figures need to be rearranged, for example, Fig.1a, b (and c?) could be supplemental figures and Fig. 1d could be one panel of Fig. 2.

Minor points:

1. In the abstract, the authors claimed “This is the first SLC4 structure, which features a fully resolved extracellular loops (EL3) and well-defined densities corresponding to sodium and carbonate ions in the tentative substrate binding pocket.” may cause confusions since it’s not the first SLC4 structure.

2. Line 103 in page 3, although molecular weight of NDCBE is about 120kDa, it forms a dimer, thus not a relatively small membrane protein for single-particle cryo-EM technique.

3. Page 16: line 541 and 542 the units of the cutoff range (may be nm) should be specified.

4. Page 18: line 606, references for CHARMM-GUI Absolute Free Energy tool and NAMD were not correctly cited.

5. Page 27: Figure 3d, how about the time series of the ion binding in the four runs? Are they initial conditions dependent? Some time series of ion positions and correlation functions between ion positions (or on/off sites status) would be suitable to reveal them.

6. Fig. S1e: please indicate which half map is used to do the refinement (work map) and which one is not used for model refinement (free map).

7. Table S1: Could the residence time in the presence of Na⁺ and Cl⁻ be also listed for comparison? Again, the prolonged time of the anion could be better understood by some kinds of correlations between the residence of the Na⁺ and Cl⁻.

8. Table S2: The binding affinity calculated for Cl⁻ only is 6.1 kCal/mol (about 10.3 kT). Why Cl⁻ is so special compared to the other anions? Is there any steric repulsions which could not be optimized?

9. Did the author include 11,824 particles (if particles are in good quality) while perform data refinement focusing on transmembrane regions? Will including more particles improve the final resolution and map quality?

Reviewer #3 (Remarks to the Author):

The manuscript of Wang et al. details the structure of rat NBCBE (SLC4A8) in the outward-facing conformation, which complements previous structures of two other SLC4 carriers, AE1 and NBCe1 (the latter from the Kurtz lab as well), also holding the same conformation. In contrast to these structures, the novel NBCBE structure shows densities in the substrate binding site. This is the first time that substrate is resolved in an SLC4 protein. Substrate binding is of particular relevance for the SLC4 family due to the polymorphism of transport modes: NBCBE catalyzes electroneutral sodium-coupled carbonate:chloride exchange, which differs from AE1 (electroneutral bicarbonate:chloride exchange) and NBCe1 (electrogenic sodium:carbonate symport). In addition the extracellular loop 3 is sufficiently well resolved to build its structure. The structural data of the paper is complemented with a computational method (SILCS) to verify the positions of the ions. In addition, molecular dynamics simulations are used to further investigate the role of the two binding sites and reveal the ion permeation paths. Finally, mutagenesis is performed as a reality-check on the proposed role of the identified residues.

The new structure that is overall better resolved and with a higher resolution is a significant achievement in itself, but regrettably does not provide new insights concerning the transport mechanism: it simply fits very well with the structures of AE1 and NBCe1. A detailed comparison between the structures is not even provided and the respective section in the discussion does not go very deep, but discusses the overall fold, the fact that the OF conformation appears most stable, and that the dimeric state and the organization of the dimer is very similar to the previous structures. These findings confirm what the previous NBCe1 structure already suggested. The fact that EL3 is resolved is very nice, but given that this loop does not appear to have a mechanistic role (apart from making the protein more susceptible to proteolysis if the disulfide bridges are incorrectly formed) this new information does not add much to our understanding of the proteins mode of operation. The computational section is interesting, though the SILCS analysis appears mostly to predict what is obvious: the anion binds near a Lys and the cation near a Asp. It would have been interesting to have had this section predict how chloride would bind in a substrate binding site that can also be occupied with sodium and carbonate. The subsequent functional studies of the identified critical residues show modest reduction of activity, mostly around 50% of the wild type protein. This appears very little and surprising given the assigned roles. Is the assay itself not sensitive enough (maybe too high substrate concentrations are used masking potential reductions in affinity?) or are these residues not relevant? Nevertheless, some of these residues are even targeted in diseases, but it is hard to imagine that a 50% reduction already leads to such a strong disease phenotype. Regrettably, no explanation is offered.

The new structure holding density of the ligands provides some insights into the transport mechanisms of SLC4 proteins, but the manuscript would have benefited from addressing these insights experimentally and in a similar way as in their previous paper on NBCe1. The computational sections are interesting, but do not offer detailed insights that could explain the different transport modes among SLC4 proteins. As such, I feel the current version of the manuscript is somewhat limited in its scope, which may be overcome by additional functional studies.

We thank the editor and reviewers for the very helpful comments. We have addressed all the issues raised in our revised manuscript. The requested text modifications in addition to clarifications are highlighted in yellow in the revised manuscript. Our responses to the reviewers' comments are detailed below.

REVIEWER COMMENTS

Reviewer #1 (Remarks to the Author):

Nature Comms 21-09497

The exciting paper from the Kurtz lab provides new insights into the structure and mode of operation of the sodium-dependent chloride/bicarbonate exchanger (NDCBE), a member of the SLC4 solute carrier family. As expected, NDCBE has an identical 14 transmembrane (TM) dimeric structure to other members of SLC4 family, featuring a 7 + 7 TM inverted topology, complex folded core and gate domains and two short opposing TM 3 and 10 helices. A novel feature revealed by this 3.5 Å cryo-EM structure is the presence of a tightly-folded glycosylated extra-cellular loop (EL3) joining TM segments 5 and 6 at the dimer interface. A second important insight is the identification of the sodium-carbonate ion pair central binding site. A third feature discovered in the structure is a more distal carbonate gathering site. The role of residues involved in substrate binding and translocation were confirmed by site-directed mutagenesis and computer modelling studies, further strengthen this fine structural study.

Comments

1. Add SLC4A8 to the title

SLC4A8 was added to the title.

2. Line 78 Include comprehensive review on SLC4A1 in reference list:

(<https://pubmed.ncbi.nlm.nih.gov/27058983/>)

This review (revised ref. 17) was added in the Introduction in the revised manuscript.

3. Line 79. Mention 7+7 fold commonly found in SLC4, SLC23 and SLC26 families with reference to:

(<https://pubmed.ncbi.nlm.nih.gov/27865089/>)

The 7+7 fold has been discussed as requested and an additional reference (revised ref. 23) was added in the Introduction of the revised manuscript.

4. Lines 87/164/282 Important to mention in the paper that the OF state was attained without use of inhibitors (or stabilizing antibody) as in the SLC4A1 structure. Line 282: "Interestingly, the finding that the structures of AE1, NBCe1, and now NDCBE were solved in the OF conformation suggests that this is the most stable conformation of the SLC4 transporters." This comparison is not valid because the structure of AE1 referenced may not be representative of its native conformation since it is 1) a crystal structure 2) of cleaved membrane domain 3) deglycosylated 4) bound to an antibody fragment and 5) labelled with H2DIDS.

The text has been revised as recommended by the reviewer (see changes in "Protein expression and purification" in Methodology and first paragraph of the Discussion section).

5. Line 108 Understand that the low resolution of the cytoplasmic domain precluded modelling. Were the authors able to determine whether the membrane and cytoplasmic domain had a cross-over domain-swap structure like SLC26A9, which is important for dimer formation: (<https://elifesciences.org/articles/46986>)?

Because the CD is dynamic/flexible, the resolution of this domain is poor compared to the rest of the dimer. As a result, we could not resolve the connection between TMD and CD sufficiently to address this question. Based on the low-resolution density map (new Fig. S1d), it appears that there is no crossover domain swap as reported by Walter and colleagues in murine Slc26a9 structure (ref. 22).

6. Line 174 The carbonate is located precisely between the N-terminal ends of TM3 and TM10. Some mention should be made of the possible role of the positive helix dipole in stabilizing anions at this site (<https://pubmed.ncbi.nlm.nih.gov/27058983/>).

This issue is now discussed in the Introduction of the revised manuscript and the reference is included (revised ref. 17).

7. Line 179 The sodium and carbonate likely form a stabilized ion pair in the S1 site accounting for the 1:1 stoichiometry. Transporters with other substrates stoichiometries may have evolved to employ different residues. E.g., The smaller Asp versus Glu in SLC4A1 can accommodate Na⁺ versus H⁺ in SLC4A1. Mention that SLC4A1 can act as a proton-sulfate co-transporter: (<https://pubmed.ncbi.nlm.nih.gov/1352774/>).

The smaller aspartate residue in NBCe1 (SLC4A4), NDCBE (SLC4A8) and other Na⁺-dependent SLC4 members instead of the glutamate present in AE1 can accommodate a larger Na⁺ ion versus an H⁺ in AE1. We originally discussed this consideration in our Huynh et al., 2018 paper on the near atomic structure of NBCe1 (ref. 11). As requested, we now refer to this issue in the Discussion of the revised manuscript. We also now refer to the Jennings and Smith 1992 paper (revised ref. 32).

8. Line 302 Some brief mention of the role of N-glycosylation should be made. Interestingly, the SLC26A2 family member also has a disulfide-stabilized tightly folded extracellular N-glycosylated domain (<https://pubmed.ncbi.nlm.nih.gov/28941661/>).

This is now mentioned in the revised Discussion section of the manuscript. Rapp et al. 2017 (revised ref. 26) was also added as recommended.

9. Line 315 While the membrane domain of SLC4A1 is a stable dimer, mention the potential role of the dimeric cytoplasmic domain, including the cross-over domain swap in SLC26A9 in Ref 21. Also mention the potential role of lipids like cholesterol in stabilizing the dimer. interface:(<https://www.sciencedirect.com/science/article/pii/S0006349519307465>).

The potential role of the CD and lipids in the dimerization is now included in the revised Discussion. In addition, De Vecchis et al. (revised ref. 30) and the cross-over domain swap in SLC26A9, mentioned in original ref. 22, are included as requested.

10. Line 324 Fig. 5 a-d is referencing the wrong figure (should be figure 6)

This is correct. Given that we moved the content of Fig. 1 to the Supplement (Fig. S1) in the revised manuscript as requested by Reviewer 2, the correct figure is now is Fig. 5.

11. Line 336 Leu 849 is not included in some of figures other than supplemental. It appears that a hydrophobic residue at this site is a feature (Fig. 6d) of sodium-dependent transporters, while exchanger

have a larger and positively-charged Arg at this site. Does solvent occupy the space or is there a structural adjustment in this area.

Leu849 was depicted in Fig. 6 (Fig. 5 in the revised manuscript) and its cysteine substituted function was provided in Fig. 5 (Fig. 4 in the revised manuscript). Its role in Na⁺ coordination was not discussed in the original manuscript since the hydrophobicity of Leu849 likely precludes it from playing a role in Na⁺ coordination. Furthermore, its replacement with the shorter and polar residue cysteine residue does not have a significant effect on NDCBE function (Fig. 5 in the original manuscript and Fig. 4 in the revised manuscript). As suggested, we now discuss this issue in the Discussion of the revised manuscript. We did not identify any solvent density in this space at the current resolution. No structural adjustment was performed in this area.

12. Thirty some references (22-59) referring to various methodological details seems excessive. Perhaps reference could be made to earlier work using similar computational methods.

The manuscript has been revised as suggested. Original references 42, 44, 45, 51-53, 57 and 58 were removed.

13. Data acquisition, lines 453-455. Based on the reported dose rate on the camera (9 e- pixel⁻¹ s⁻¹), the total exposure time (8s) and reported pixel size (0.535 Å), the calculated total electron exposure is 251.55 e-/Å² (9 * 8 / (0.535²)), which is inconsistent with that reported in Table 1 (52 e-/Å²).

*The aligned micrographs with dose-weighting were generated with bin 2 (corresponding pixel size of 1.07 Å (0.535*2)). The dose rate was set to 8.6 e- pixel⁻¹ s⁻¹ at the beginning, but we found that the dose rate decreased to ~7 e- pixel⁻¹ s⁻¹ at the end of collection. An estimated frame dose rate of 1.3 e-/Å² was used for motion correction. Therefore, the total electron dose is 1.3 e-/Å²/frame*40 frame=52 e-/Å². We corrected the dose rate to 7.9 e- pixel⁻¹ s⁻¹ in the data acquisition section so that it is consistent with that of Table 1; 52 (7.9*8 / (1.1²)).*

14. Clashscore of 16.3 reported in Table 1 is high, indicating a large number of clashes in the model. In our experience, automated refinement using PHENIX in this resolution range is not sufficient to resolve these issues. We recommend manual correction of these issues, or automated correction using Rosetta FastRelax or ISOLDE. A clashscore of approx. 1-2 is certainly achievable at this resolution.

The final Clashscore is 1.4 after manual correction. The manuscript has been revised accordingly (see Table 1).

15. The distances reported in Figure 3b between T538 and the carbonate anion are 2.2 and 2.4Å, shorter than the 2.8Å typically found in protein structures. In contrast, the distances between D800, T847, T804 and the sodium ion are greater than expected for sodium coordination (normally 2.42 Å, MM Harding Acta Cryst. (2002). D58, 872-874). Some of these residues could be indirectly coordinating the sodium ion by coordinating water molecules (which may not be visible in the map at 3.0Å resolution). The distances described in this figure may change following refinement with other software packages described above.

To evaluate the signal of ions difference map, the experimental EM density map minus simulated map from the refined model without ions was calculated. The distances between ions and coordinating residues were adjusted according to Acta Cryst. (2002) D58, 872-874 (revised ref. 41). However, the distances between D800, T847 and the Na⁺ ion are 2.6 Å. This indicates that some of these residues may coordinate Na⁺ indirectly via water molecules as suggested by the reviewer. This is now discussed in the Results section of the revised manuscript.

16. The density for D800 in Figure 3c is not shown. Was this omitted or is the density not strong enough (due to radiation damage, which affects negatively charged residues)?

The density for D800 sidechain was not shown because it was not observed well possibly because of the flexibility of the sidechain or due to radiation damage for a negatively charged residue, as the reviewer suggests.

17. Orientation distribution should be shown in Figure S1.

Orientation distribution was added to the revised Fig. S1 as requested.

18. EMRinger test for side-chain (available in Phenix) should be used, and results reported in Table 1.

EMRinger score (2.10) was added to the revised Table 1 as requested.

Reviewer #2 (Remarks to the Author):

The manuscript by Wang and colleagues reported the structure of rat NDCBE (SLC4A8), a Na⁺-driven Cl⁻/CO₃²⁻ exchanger at near-atomic resolution using single-particle cryo-EM. Similar to previously determined structures in SLC4 family, NDCBE forms a dimer and the overall structure shows an outward facing conformation. Based on structural analysis, the putative ion (Na⁺, CO₃²⁻) binding sites have been identified and further confirmed by molecular dynamic simulation and mutagenesis. One major point of this manuscript is identification of the ion binding sites for the first time in SLC4 family. However, the figure showing in the current version (fig. 3C) is difficult to evaluate the quality.

The authors can calculate the difference map, i.e. experimental EM density map minus simulated map from refined atomic model without interested ion(s). The omit density can be used to evaluate the signal of the built ions. Also, the modeled Na⁺ and CO₃²⁻ are very close, could corresponding ion densities be accurately assigned confidently?

As discussed above (Response to Reviewer 1, comment 15), the difference map was calculated by using the experimental EM density map minus the simulated map from the model without ions. The distances between ions and coordinating residues were adjusted in the revised Fig. 2.

In line 170 of page 5, the authors claimed they modeled Na⁺ because that's the only cation they used in buffer. The possibility of endogenous ions binding to NDCBE tightly from protein expression can't be excluded.

We agree that this possibility cannot be absolutely excluded, however we consider it is unlikely, given that NDCBE has a functional preference for Na⁺, indicating that its affinity for Na⁺ in the ion coordination site is likely preferential to K⁺, Ca²⁺ and Mg²⁺. In addition, the fact that Na⁺ was the only cellular cation in our protein purification buffer that the transporter was exposed to for a long time (~30 hours) essentially rules out the possibility that other ions, which were absent in the buffer, are still binding to the transporter.

In line 281 of page 9, the authors made conclusion that "... NDCBE were solved in the OF conformation suggests that this is the most stable conformation of the SLC4 transporters". Although current structures available in SLC4 family all present an OF conformation, this statement needs more evidence to support.

We agree that it is currently a hypothesis and we have removed it from the revised manuscript.

In line 320-322 of page 10, the authors said "Therefore, the transport mode of a SLC4 transporter is determined by a small number of residues in/near the ion coordination area rather than the overall 3D fold." First, even this statement is correct, 3D structure of the protein is the fundamental basis that allows the arrangement of these key residues in a way to play their important roles. Secondly, current

evidence is not solid and strong to support this statement. The author need experimental data to show that not only functions are switched by replacing key residues, but other features for transporter and exchangers in SLC4 family, like ions selectivity and specificity, are also maintained as well as the wide type protein.

This consideration originally appeared in our Huynh et al. 2018 paper on the near atomic resolution structure of NBCe1, in which replacement of several amino acids in NBCe1 with corresponding AE1 residues allowed NBCe1 to function in a Cl⁻-driven transport mode. We certainly agree with the reviewer that the overall structure of the transporters plays a very important role in their function but what we wanted to emphasize is that in a structural context, the residues in the ion coordination site significantly affect the transport mode. We have now modified the manuscript.

The manuscript should be revised to make a more smooth manuscript. For example, description of the cryo-EM analysis (the first part of Results section in page 3) contains technical details that are redundant with the one in methods section.

This part could be integrated into other structural analysis part, like overall structure (Page 4) and/or NDCBE dimerization (Page 5) section. ?

Similar redundant descriptions is found in the discussion part that already exist in Results part.

Also, the figures need to be rearranged, for example, Fig.1a, b (and c?) could be supplemental figures and Fig. 1d could be one panel of Fig. 2.

We have revised the manuscript as suggested. Specifically, the first part of the Results section is now integrated into the overall structure section. Fig. 1a-c are now integrated into the revised Fig. S1. The redundancies have been removed as suggested.

Minor points:

1. In the abstract, the authors claimed “This is the first SLC4 structure, which features a fully resolved extracellular loops (EL3) and well-defined densities corresponding to sodium and carbonate ions in the tentative substrate binding pocket.” may cause confusions since it’s not the first SLC4 structure.

We had originally written in the original submission: “This is the first SLC4 structure, which features a fully resolved extracellular loop 3 (EL3) and well-defined densities corresponding to sodium and carbonate ions in the tentative substrate binding pocket.” What we were emphasizing is that NDCBE is the first SLC4 structure, which features a fully resolved EL3 (the largest EL) and well-defined densities corresponding to Na⁺ and CO₃²⁻ in the tentative substrate binding pocket not resolved in previous studies. We feel that the sentence in the original submission does not give the impression that this is the first SLC4 transporter structure solved at near atomic resolution but rather this is the first SLC4 structure, which features a fully resolved EL3 etc.

2. Line 103 in page 3, although molecular weight of NDCBE is about 120kDa, it forms a dimer, thus not a relatively small membrane protein for single-particle cryo-EM technique.

We have removed this statement in the revised manuscript.

3. Page 16: line 541 and 542 the units of the cutoff range (may be nm) should be specified.

The units (Å) for the cutoff ranges used in the SILCS calculations are now provided in the Methods section of the revised manuscript.

4. Page 18: line 606, references for CHARMM-GUI Absolute Free Energy tool and NAMD were not correctly cited.

We have corrected the citations as requested.

5. Page 27: Figure 3d, how about the time series of the ion binding in the four runs? Are they initial conditions dependent? Some time series of ion positions and correlation functions between ion positions (or on/off sites status) would be suitable to reveal them.

We have provided new figures (revised Figs. S5 and S6) in the SI section in the revised manuscript. They now show time series of ion residence in the different areas of the OF cavity and correlation between the Na^+ and CO_3^{2-} ions bound initially as an ion pair in the area of sites S1/S2 extracted from the four 1 μs MD simulations (Trials 1-4 in Table S1) as suggested by the reviewer. We also provide relevant residence time information in Table S1. All four 1 μs ANTON runs were seeded from distinct coordinates/velocities extracted from unconstrained MD simulations performed with NAMD as described in the revised Methods section and featured different initial positions of the Na^+ - CO_3^{2-} pair in the S1/S2 binding pocket. We have also made additional edits throughout the main text to reflect and cite this newly provided data.

6. Fig. S1e: please indicate which half map is used to do the refinement (work map) and which one is not used for model refinement (free map).

Half map 1 was used for the refinement. The full map was used for model refinement. This information has been added in the Methods section (Image processing section).

7. Table S1: Could the residence time in the presence of Na^+ and Cl^- be also listed for comparison? Again, the prolonged time of the anion could be better understood by some kinds of correlations between the residence of the Na^+ and Cl^- .

Table S1 now shows the residence time of the initially bound ions including Na^+ and Cl^- , specifically in the area of the S1/S2 binding pocket evaluated from the new time series plots for all trajectories listed in Table S1, presented in Fig. S5 and S7. The behavior of the ions in the 300 ns MD trajectories is similar to the one observed in the 1 μs trajectories (Figs. S5, S7). The initially bound ions eventually dissociate from the binding pocket and leave the cavity. When the systems are loaded with a Na^+ -anion pair, they tend to dissociate together. This is evident from the plots in Fig. S7 and the Na^+ and anion residence times in Table S1, which are the same for the two counterions from the initially loaded ion-pair. The presence of Na^+ leads to longer anion residence. Occasional short-lived entry of Na^+ or Cl^- ions from the solutions in the binding pocket is also observed.

8. Table S2: The binding affinity calculated for Cl^- only is 6.1 kcal/mol (about 10.3 kT). Why Cl^- is so special compared to the other anions? Is there any steric repulsions which could not be optimized?

The analysis of the free energy calculations done specifically for the site S1 in the current OF state provides some insights into the key drivers of differences between anions. In this site, in the absence of the anchoring Na^+ , a trigonal planar divalent anion like CO_3^{2-} interacts far strongly with the NH-groups of the V844-V848 stretch and the side chain of the nearby K970 than a spherical monovalent anion such as Cl^- . The coordination mode involving backbone amide hydrogens and lysine side chain has been shown to be an essential driver of selectivity for oxoanions in the model peptides (Milner-White et al. Acta Crystallographica, Section D: Biol. Crystallography, 60, 1935-1942, 2004). The spherical Cl^- is still preferentially coordinated by water molecules with only a few protein contacts. The monovalent HCO_3^- also seems to bind more strongly to the site S1 via additional stabilization from the V844-V848 motif. We have discussed this in the "Evidence for Na^+ -dependent anion binding in site S1 from binding free energy calculations" section of Results.

9. Did the author include 11,824 particles (if particles are in good quality) while performing data refinement focusing on transmembrane regions? Will including more particles improve the final resolution and map quality?

11,824 particles were used for the full-length structure refinement including TMD and CD. 380,776 particles were included while performing data refinement focusing on transmembrane regions. Use of larger numbers of particles did not improve the final resolution probably because not all particles were of sufficient quality.

Reviewer #3 (Remarks to the Author):

The manuscript of Wang et al. details the structure of rat NBCBE (SLC4A8) in the outward-facing conformation, which complements previous structures of two other SLC4 carriers, AE1 and NBCe1 (the latter from the Kurtz lab as well), also holding the same conformation. In contrast to these structures, the novel NBCBE structure shows densities in the substrate binding site. This is the first time that substrate is resolved in an SLC4 protein. Substrate binding is of particular relevance for the SLC4 family due to the polymorphism of transport modes: NBCBE catalyzes electroneutral sodium-coupled carbonate:chloride exchange, which differs from AE1 (electroneutral bicarbonate:chloride exchange) and NBCe1 (electrogenic sodium:carbonate symport). In addition the extracellular loop 3 is sufficiently well resolved to build its structure. The structural data of the paper is complemented with a computational method (SILCS) to verify the positions of the ions. In addition, molecular dynamics simulations are used to further investigate the role of the two binding sites and reveal the ion permeation paths. Finally, mutagenesis is performed as a reality-check on the proposed role of the identified residues.

The new structure that is overall better resolved and with a higher resolution is a significant achievement in itself, but regrettably does not provide new insights concerning the transport mechanism: it simply fits very well with the structures of AE1 and NBCe1.

The transport mechanism of an ion transporter is a large field in and of itself and involves a number of aspects, which include ion binding site(s), ion permeation, ion specificity, stoichiometry, kinetics, transport rates, utilization of electrochemical gradients, substrate affinities, coupling and binding cooperativity between substrates, thermodynamics, conformational changes, effect of bilayer lipids, effect of membrane potential and others. Our structural and functional data were readily used to address certain aspects of the transport mechanism focusing on the number and location of the substrate binding sites. Importantly, prior to our study, the precise location and number of ion binding sites was unclear and was hypothesized based on functional studies and general structural considerations. Based on our MD simulations complemented by our structural and functional data, we have discovered that there are 2 ion binding sites in NDCBE, peripheral and central. Importantly, our study is also the first that identified the residues involved in Na⁺ and anion interaction in the binding pocket and along the permeation pathways, which contribute to the observed SLC4 function. These findings are novel and provide important new insights into key aspects of the mechanism of Na⁺-dependence in the SLC4 transport. Furthermore, by comparing with AE1 and NBCe1, our data provide an important new framework to address the distinct functional properties of SLC4 transporters in general in future studies.

A detailed comparison between the structures is not even provided and the respective section in the discussion does not go very deep, but discusses the overall fold, the fact that the OF conformation

appears most stable, and that the dimeric state and the organization of the dimer is very similar to the previous structures. These findings confirm what the previous NBCe1 structure already suggested.

We have improved the comparison between NDCBE, AE1 and NBCe1 structures in the Discussion section of the revised manuscript as suggested. We use this comparison to discuss potential mechanistic implications in the three proteins areas of sites S1/S2 and along TM3 and TM5.

The fact that EL3 is resolved is very nice, but given that this loop does not appear to have a mechanistic role (apart from making the protein more susceptible to proteolysis if the disulfide bridges are incorrectly formed) this new information does not add much to our understanding of the proteins mode of operation.

The following are our considerations regarding the importance in the field of resolving EL3 for the first time in an SLC4 transporter: First, EL3 is the largest EL and is a characteristic feature of Na⁺-dependent SLC4 transporters. It has 2 putative glycosylation sites, both linked to sugars in NDCBE. In contrast, AE1 (and presumably AE2 and AE3) has a single putative glycosylation site located in significantly shorter EL3. Khandoudi et al. (Cardiovasc. Res. 52, 387–396, 2001) reported that an antibody targeted to EL3 of another Na⁺-dependent SLC4 transporter, NBCe1, provided substantial protection to rat hearts subjected to ischemia and reperfusion, suggesting a potential functional role in this regard. This point has been added to the revised manuscript. Secondly, we have demonstrated for the first time in this paper that EL3 is located at the dimeric interface between the two subunits and is therefore structurally involved in dimerization (see revised Fig. 1 and NDCBE dimerization section). This is now mentioned in the Results and Discussion sections of the revised manuscript.

The computational section is interesting, though the SILCS analysis appears mostly to predict what is obvious: the anion binds near a Lys and the cation near a Asp. It would have been interesting to have had this section predict how chloride would bind in a substrate binding site that can also be occupied with sodium and carbonate.

We agree with the Reviewer that we are currently limited in our ability to provide a full explanation of Cl⁻ binding in NDCBE. SILCS methodology is yet to be expanded for modelling of small ions directly such as Cl⁻ as it uses only polyatomic fragments as probes. We beg to disagree with the Reviewer on the obviousness of SILCS data, as the maps collected are revealing dominant entry pathways for anions and cations in a protein with MULTIPLE basic and acidic residues and a multitude of possible permeation routes. We are working in a collaboration with Alexander MacKerell, SILCS developer, to establish a set of excess chemical potentials needed to calculate the mobile cation and anion maps and then will use them for further evaluation of ion specificity in the SLC4 family. This remains a future and very promising direction for SILCS applications to membrane proteins, as noted by the reviewer. However, our current study combines multi-microsecond all-atom MD and Free Energy simulations to clearly demonstrate that site S1 in the OF state (mapped by Cryo-EM density and SILCS maps) has lower affinity for Cl⁻, than for the other studied anions. As we noted in our answer to Reviewer 2, FEP calculations, albeit limited to analysis of only site S1 in OF conformation suggest that anion geometry and matching protein coordination appear to be important factors in anion specificity of this proteins. Indeed, Cl⁻ might bind more strongly in a different state. Ion competition due to the significantly larger concentration of Cl⁻ compared to CO₃²⁻ in the extracellular space might also play a role. Kinetics and thermodynamics of ion permeation, which might be affected by environmental factors (e.g. pH) could lead to changes in transport efficiency for different ions. We believe that our study already provides some critical for the field novel findings (see comments above), including the fact that Cl⁻ does not seem to bind as strongly to site S1 in NDCBE as other anions, which merits further studies. The explanation of the full transport

activity of the SLC4 proteins would require resolution of additional conformations from their catalytic cycle and additional computational and functional work, which is outside of the scope of the present studies. The Discussion has now been revised accordingly to address these considerations.

The subsequent functional studies of the identified critical residues show modest reduction of activity, mostly around 50% of the wild type protein. This appears very little and surprising given the assigned roles. Is the assay itself not sensitive enough (maybe too high substrate concentrations are used masking potential reductions in affinity?) or are these residues not relevant? Nevertheless, some of these residues are even targeted in diseases, but it is hard to imagine that a 50% reduction already leads to such a strong disease phenotype. Regrettably, no explanation is offered.

There are a number of previous papers in the literature, which have studied mutant proteins causing a disease phenotype where the function or protein amount is decreased by ~50%. In these studies, an ~50% decrease in function or expression is considered highly significant because it was sufficient to cause disease. Some of the diseases caused by loss of ~50% of protein function or expression are CHARGE syndrome, Cleidocranial dysostosis, Ehlers–Danlos syndrome, Frontotemporal dementia, De Vivo disease, Holoprosencephaly, Holt–Oram syndrome, Marfan syndrome, Phelan–McDermid syndrome, Preaxial polydactyly and Dravet syndrome. The magnitude of the functional decrease of several of the single mutants in our study is comparable to the decrement in activity or expression reported in the literature that were considered significant. Literature references include: J Biol Chem 273, 6380–6388 (1998), Hum Mol Genet 8, 2311–2316 (1999), Am J Hum Genet 66, 1766–1776 (2000), J Clin Invest 114, 172–181 (200), J Med Genet 41, e113 (2004), Hum Mutat 29, 512–521 (2008), Developmental Cell 15, 236–247 (2008), Genetics in Medicine 13, 717–722, (2011), Hum Mol Genet 21, 1888–1896 (2012), Epilepsia 53, 87–100 (2012), Disease Models Mechanisms 7, 535–545 (2014), J Immunol (194), 2190–2198 (2015), Human Genome Variation 3, 1–3 (2016), and Developmental Cell 56, 292–309 (2021). Secondly, the lack of a greater functional inhibition using the single residue mutants in our study is expected because more than a single residue is involved in the interaction with Na^+ and CO_3^{2-} . Each of these ions is coordinated by several residues and therefore when a single residue is mutated, ion coordination can still occur but to a lesser extent. This is clearly shown in the additional mutant functional studies we have now performed in the revised manuscript, where more than one residue was mutated at a time showing greater inhibition when compared to a single mutant inhibition. Thirdly, from a technical aspect the functional assay we used has been utilized in the literature by multiple authors over many years and is extremely sensitive and reproducible. It favored by most authors since it can detect small changes in function reliably and reproducibly. The typical substrate concentration changes (physiologic to zero) we and other investigators in the field use to drive the transporter activity are such that if there is a decrement in the affinity of the transporter due to a mutation, this will manifest in a change in the rate of flux that is measured in the assay. Fourthly, as suggested by the reviewer, we have performed a number of additional experiments utilizing multiple combinations of the cysteine mutants reported in the original submission. These results combined with the original data which are presented in the revised Figs. 4 and S8. complement and confirm our previous findings demonstrating the key residues for transport.

The new structure holding density of the ligands provides some insights into the transport mechanisms of SLC4 proteins, but the manuscript would have benefited from addressing these insights experimentally and in a similar way as in their previous paper on NBCe1.

Please see our previous comment on the transport mechanism. Furthermore, one of the approaches taken in this study as in our NBCe1 study (ref. 11) was to choose key residues that were mutated and functionally studied. In the present study, specific residues were chosen based on our structural data and the location of substrates in the transporter, in addition to our MD analyses. In the NBCe1 paper, only the structural data was utilized since we didn't have the substrate localization information or the MD analyses both of which represent a significant advance in our understanding of NDCBE (and SLC4) biology.

The computational sections are interesting, but do not offer detailed insights that could explain the different transport modes among SLC4 proteins. As such, I feel the current version of the manuscript is somewhat limited in its scope, which may be overcome by additional functional studies.

If by transport mode the reviewer refers to exchanger versus cotransporter, our work on NBCe1 provided initial insight into mode of transport considerations by showing that residues in the ion coordination site of NBCe1 contribute as discussed above. A complete understanding of the differences in mode of transport will likely require in the future that the IF conformation of NDCBE and other SLC4 transporters be structurally determined. Thus far, no one has succeeded in characterizing such a structure. As suggested by the reviewer, we have performed several additional functional studies, which are now included in the revised manuscript (see the response above).

REVIEWERS' COMMENTS

Reviewer #2 (Remarks to the Author):

The authors have addressed most of my concerns raised in the previous review. The current version of paper is further strengthened and nearly ready for publication. Prior to publication, the authors should consider a few minor comments below and modify the manuscript if necessary.

Two minor comments:

1. Line 448: Please indicate grid type used in this study, copper, gold or others.
2. Line 452: Reference for SerialEM is missing.
3. Fig. S1C: From 7545 movie stacks to 7393 micrographs, consider adding manual inspection together with motion correction.

Reviewer #3 (Remarks to the Author):

The main finding of the authors, a near-atomic description of the liganded NBCBE substrate binding site, is better represented in the revised manuscript. Among others, this concerns the more detailed comparison with the already available structures of AE1 and NBCe1. Furthermore, the additional functional studies demonstrate a (surprisingly) strong redundancy in the substrate binding site, where mutations of multiple residues are required to reach a reduction in transport exceeding 40-50%.

The authors have appropriately addressed my comments.

Minor remarks

line242/238: Pro540 is indicated as being part of S1/S2 and as being in the vicinity of S1/S2. Clearly, both is not possible. Please correct.

line 355: protonation of Asp800 is suggested as a possible strategy to allow chloride binding. The authors need to explain how protonation in the presence of chloride (combined with a deprotonated state in the presence of Na⁺ and carbonate) would reconcile with the fact that SLC4e8 mediates the electroneutral exchange of sodium-carbonate with chloride. Clearly, having a proton co-transported with the chloride ion would lead to the net import of a negative charge per translocation cycle. Please explain or remove this suggestion.

We thank the editor and reviewers for the very helpful comments. We have addressed the minor issues remaining and have made the modifications in our revised manuscript.

REVIEWER COMMENTS

Reviewer #2 (Remarks to the Author):

The authors have addressed most of my concerns raised in the previous review. The current version of paper is further strengthened and nearly ready for publication. Prior to publication, the authors should consider a few minor comments below and modify the manuscript if necessary.

Two minor comments:

1. Line 448: Please indicate grid type used in this study, copper, gold or others.
2. Line 452: Reference for SerialEM is missing.
3. Fig. S1C: From 7545 movie stacks to 7393 micrographs, consider adding manual inspection together with motion correction.

The manuscript has been modified accordingly.

Reviewer #3 (Remarks to Author):

The main finding of the authors, a near-atomic description of the liganded NBCBE substrate binding site, is better represented in the revised manuscript. Among others, this concerns the more detailed comparison with the already available structures of AE1 and NBCe1. Furthermore, the additional functional studies demonstrate a (surprisingly) strong redundancy in the substrate binding site, where mutations of multiple residues are required to reach a reduction in transport exceeding 40-50%.

The authors have appropriately addressed my comments.

Minor remarks

line242/238: Pro540 is indicated as being part of S1/S2 and as being in the vicinity of S1/S2. Clearly, both is not possible. Please correct.

line 355: protonation of Asp800 is suggested as a possible strategy to allow chloride binding. The authors need to explain how protonation in the presence of chloride (combined with a deprotonated state in the presence of Na⁺ and carbonate) would reconcile with the fact that SLC4e8 mediates the electroneutral exchange of sodium-carbonate with chloride. Clearly, having a proton co-transported with the chloride ion would lead to the net import of a negative charge per translocation cycle. Please explain or remove this suggestion.

The manuscript has been modified accordingly.